# Intrinsically Efficient, Stable, and Bounded Off-Policy Evaluation for Reinforcement Learning

**Nathan Kallus**
Cornell University
New York, NY
kallus@cornell.edu

**Masatoshi Uehara** *
Harvard University
Cambrdige, MA
uehara_m@g.harvard.edu

## Abstract

Off-policy evaluation (OPE) in both contextual bandits and reinforcement learning allows one to evaluate novel decision policies without needing to conduct exploration, which is often costly or otherwise infeasible. The problem's importance has attracted many proposed solutions, including importance sampling (IS), self-normalized IS (SNIS), and doubly robust (DR) estimates. DR and its variants ensure semiparametric local efficiency if Q-functions are well-specified, but if they are not they can be worse than both IS and SNIS. It also does not enjoy SNIS's inherent stability and boundedness. We propose new estimators for OPE based on empirical likelihood that are always more efficient than IS, SNIS, and DR and satisfy the same stability and boundedness properties as SNIS. On the way, we categorize various properties and classify existing estimators by them. Besides the theoretical guarantees, empirical studies suggest the new estimators provide advantages.

## 1 Introduction

Off-policy evaluation (OPE) is the problem of evaluating a given policy (evaluation policy) using data generated by the log of another policy (behavior policy). OPE is a key problem in both reinforcement learning (RL) [7, 13–15, 17, 23, 32] and contextual bandits (CB) [5, 19, 28] and it finds applications as varied as healthcare [18] and education [16].

Methods for OPE can be roughly categorized into three types. The first approach is the *direct method* (DM), wherein we directly estimate the Q-function using regression and use it to directly estimate the value of the evaluation policy. The problem of this approach is that if the model is wrong (misspecified), the estimator is no longer consistent.

The second approach is importance sampling (IS; aka Horvitz-Thompson), which averages the data weighted by the density ratio of the evaluation and behavior policies. Although IS gives an unbiased and consistent estimate, its variance tends to be large. Therefore self-normalized IS (SNIS; aka Hájek) is often used [28], which divides IS by the average of density ratios. SNIS has two important properties: (1) its value is bounded in the support of rewards and (2) its conditional variance given action and state is bounded by the conditional variance of the rewards. This leads to increased stability compared with IS, especially when the density ratios are highly variable due to low overlap.

The third approach is the doubly robust (DR) method, which combines DM and IS and is given by adding the estimated Q-function as a control variate [5, 7, 25]. If the Q-function is well specified, DR is locally efficient in the sense that its asymptotic MSE achieves the semiparametric lower bound [33]. However, if the Q-function is misspecified, DR can actually have *worse* MSE than IS and/or SNIS [12]. In addition, it does not have the boundedness property.

To address these deficiencies, we propose novel OPE estimators for both CB and RL that are guaranteed to improve over both (SN)IS and DR in terms of asymptotic MSE (termed *intrinsic*

Table 1: Comparison of policy evaluation methods. The notation (*) means proposed estimator. The notation # means partially satisfied, as discussed in the text. (S)IS and SN(S)IS refer either to stepwise or non-stepwise.

| | DM | (S)IS | SN(S)IS | DR | SNDR | MDR | REG(*) | SNREG(*) | EMP(*) |
|---|---|---|---|---|---|---|---|---|---|
| Consistency | | ✓ | ✓ | ✓ | ✓ | ✓ | ✓ | ✓ | ✓ |
| Local efficiency | ✓ | | | ✓ | ✓ | ✓ | ✓ | ✓ | ✓ |
| Intrinsic efficiency | | | | | | # | ✓ | # | ✓ |
| Boundedness | 1 | | 1 | | 2 | | | 2 | 1 |
| Stability | # | | ✓ | | # | | | | ✓ |

*efficiency*) and at the same time also satisfy the same boundedness and stability properties of SNIS, in addition to the consistency and local efficiency of existing DR methods. See Table 1. Our general strategy to obtain these estimators is to (1) make a parametrized class of estimators that includes IS, SNIS, and DR and (2) choose the parameter using either a regression way (REG) or an empirical likelihood way (EMP). The benefit of these new properties in practice is confirmed by experiments in both CB and RL settings.

## 2 Sequential Decision Processes and Off Policy Evaluation

A sequential decision process is defined by a tuple $(\mathcal{X}, \mathcal{A}, P, R, P_0, \gamma)$, where $\mathcal{X}$ and $\mathcal{A}$ are the state and action spaces, $P_r(x, a)$ is the distribution of the bounded random variable $r(x, a) \in [0, R_{\max}]$ being the immediate reward of taking action $a$ in state $x$, $P(\cdot|x, a)$ is the transition probability distribution, $P_0$ is the initial state distribution, and $\gamma \in [0, 1]$ is the discounting factor. A policy $\pi : \mathcal{X} \times \mathcal{A} \to [0, 1]$ assigns each state $x \in \mathcal{X}$ a distribution over actions with $\pi(a|x)$ being the probability of taking actions $a$ into $x$. We denote $\mathcal{H}_{T-1} = (x_0, a_0, r_0, \cdots, x_{T-1}, a_{T-1}, r_{T-1})$ as a $T$-step trajectory generated by policy $\pi$, and define $R_{T-1}(\mathcal{H}_{T-1}) = \sum_{t=0}^{T-1} \gamma^t r_t$, which is the return of trajectory. Our task is to estimate

$$\beta_T^\pi = \mathrm{E}[R_{T-1}(\mathcal{H}_{T-1})] \qquad \text{(policy value)}.$$

We further define the value function $V^\pi(x)$ and Q-function $Q^\pi(x, a)$ of a policy $\pi$, respectively, as the expectation of the return of a $T$-step trajectory generated by starting at state $x$ and state-action pair $(x, a)$. Note that the contextual bandit setting is a special case when $T = 1$.

The off-policy evaluation (OPE) problem is to estimate $\beta^* = \beta_T^{\pi_e}$ for the evaluation policy $\pi_e$ from $n$ observation of $T$-step trajectories $\mathcal{D} = \{\mathcal{H}_{T-1}^{(i)}\}_{i=1}^n$ independently generated by the behavior policy $\pi_b$. Here, we assume an overlap condition: for all state-action pair $(x, a) \in \mathcal{X} \times \mathcal{A}$ if $\pi_b(a|x) = 0$ then $\pi_e(a|x) = 0$. Throughout, expectations $\mathrm{E}[\cdot]$ are taken with respect to a behavior policy. For any function of the trajectory, we let

$$\mathrm{E}_n[f(\mathcal{H}_{T-1})] = n^{-1} \sum_{i=1}^n f(\mathcal{H}_{T-1}^{(i)}).$$

Asmse$[\cdot]$ denotes asymptotic MSE in terms of the first order; *i.e.*, Asmse$[\hat{\beta}] = \mathrm{MSE}[\hat{\beta}] + \mathrm{o}(n^{-1})$.

The cumulative importance ratio from time step $t_1$ to time step $t_2$ is

$$\omega_{t_1:t_2} = \prod_{t=t_1}^{t_2} \pi_e(a_t|x_t)/\pi_b(a_t|x_t),$$

where the empty product is 1. We assume that this weight is bounded for simplicity.

### 2.1 Existing Estimators and Properties

We summarize three types of estimators. Some estimators depend on a model $q(x, a; \tau)$ with parameter $\tau \in \Theta_\tau$ for the Q-function $Q^{\pi_e}(x, a)$. We say the model is correct or well-specified if there is some $\tau_0$ such that $Q^{\pi_e}(x, a) = q(x, a; \tau_0)$ and otherwise we say it is wrong or misspecified. Throughout, we make the following assumption about the model

**Assumption 2.1.** *(a1)* $\Theta_\tau$ *is compact, (a2)* $|q(x, a; \tau)| \le R_{\max}$.

**Direct estimator:** DM is given by fitting $\hat{\tau}$, *e.g.*, by least squares, and then plugging this into

$$\hat{\beta}_{\mathrm{dm}} = \mathrm{E}_n \left[ \sum_{a \in \mathcal{A}} \pi_e(a|x_0^{(i)}) q(x_0^{(i)}, a; \hat{\tau}) \right]. \tag{1}$$

When this model is correct, $\hat{\beta}_{\mathrm{dm}}$ is both consistent for $\beta^*$ and locally efficient in that its asymptotic MSE is minimized among the class of all estimators consistent for $\beta^*$ [19, 33].

**Definition 2.1** (Local efficiency). *When the model $q(x, a; \tau)$ is well-specified, the estimator achieves the efficiency bound.*

However, all of models are wrong to some extent. In this sense, even if the sample size goes to infinity, $\hat{\beta}_{\mathrm{dm}}$ might not be consistent.

**Definition 2.2** (Consistency). *The estimator is consistent for $\beta^*$ irrespective of model specification.*

**Importance sampling estimators:** IS and step-wise IS (SIS) are defined respectively as

$$\hat{\beta}_{\mathrm{is}} = \mathrm{E}_n \left[ \omega_{0:T-1} \sum_{t=0}^{T-1} \gamma^t r_t \right], \; \hat{\beta}_{\mathrm{sis}} = \mathrm{E}_n \left[ \sum_{t=0}^{T-1} \omega_{0:t} \gamma^t r_t \right].$$

The weights $\omega_{0:t}$ are assumed known here as is common in RL; otherwise they can either be estimated directly or chosen by optimal balance [1, 8, 9]. Both IS and SIS satisfy consistency but the MSE of SIS estimator is smaller than regular IS estimator by the law of total variance [27]. The self-normalized versions of these estimators are:

$$\hat{\beta}_{\mathrm{snis}} = \frac{\mathrm{E}_n \left[ \omega_{0:T-1} \sum_{t=0}^{T-1} \gamma^t r_t \right]}{\mathrm{E}_n[\omega_{0:T-1}]}, \; \hat{\beta}_{\mathrm{snsis}} = \mathrm{E}_n \left[ \sum_{t=0}^{T-1} \frac{\omega_{0:t}}{\mathrm{E}_n[\omega_{0:t}]} \gamma^t r_t \right].$$

SN(S)IS have two advantages over (S)IS. First, they are both 1-bounded in that they are bounded by the theoretical upper bound of reward.

EMP = REG = DR
⌢
IS, SNIS

(a) Well-specified

EMP = REG
⌢
IS, SNIS, DR

(b) Misspecified

Figure 1: Order of asymptotic MSEs

**Definition 2.3** ($\alpha$-Boundedness). *The estimator is bounded by $\alpha \sum_{t=0}^{T-1} \gamma^t R_{\max}$.*

1-boundedness is the best we can achieve where $\alpha$-boundedness for any $\alpha > 1$ is a weaker property. Second, their conditional variance given state and action data are no larger than the conditional variance of any reward, to which we refer as *stability*.

**Definition 2.4** (Stability). *Let $\mathcal{D}_{x,a} = \{(x_t^{(i)}, a_t^{(i)}) : i \le n, t \le T-1\}$ denote that action-state data. If the conditional variance of $\sum_{t=1}^{T-1} \gamma^t r_t^{(i)}$, given $\mathcal{D}_{x,a}$, is bounded by $\sigma^2$, then the conditional variance of the estimator, given $\mathcal{D}_{x,a}$, is also bounded by $\sigma^2$.*

Unlike efficiency, boundedness and stability are finite-sample properties. Notably (S)IS lacks both of these properties, which explains its unstable performance in practice, especially when density ratios can be very large. While boundedness can be achieved by a simple truncation, stability cannot.

**Doubly robust estimators:** A DR estimator for RL [7, 32] is given by fitting $\hat{\tau}$ and plugging it into

$$\hat{\beta}_{\mathrm{dr}} = \hat{\beta}_d(\{q(x, a; \hat{\tau})\}_{t=0}^{T-1}),$$

where for any collection of functions $\{m_t\}_{t=0}^{T-1}$ (known as control variates) we let

$$\hat{\beta}_d(\{m_t\}_{t=0}^{T-1}) = \mathrm{E}_n \left[ \sum_{t=0}^{T-1} \gamma^t \omega_{0:t} r_t - \gamma^t \left( \omega_{0:t} m_t(x_t, a_t) - \omega_{0:t-1} \left( \sum_{a \in A} m_t(x_t, a) \pi_e(a|x_t) \right) \right) \right]. \tag{2}$$

The DR estimator is both consistent and locally efficient. Recently, this approach to efficient OPE has been extended to Markov decision process [10] and infinite-horizon problems [11]; in this paper we focus on the more general sequential decision process. Instead of using a plug-in estimate of $\tau$ in eq. 2, [4, 6, 26] further suggest that to pick $\hat{\tau}$ to minimize an estimate of the asymptotic variance of $\hat{\beta}_d(\{q(x, a; \tau)\}_{t=0}^{T-1})$, leading to the MDR estimator [6] for OPE. However, DR and MDR satisfy neither boundedness nor stability. Replacing, $\omega_{0:t}$ with its self-normalized version $\omega_{0:t}/\mathrm{E}_n[\omega_{0:t}]$ in (2) leads to SNDR [24, 32] (aka WDR), but it only satisfies these properties partially: it's only 2-bounded and partially stable (see Appendix B).

Moreover, if the model is incorrectly specified, (M)DR may have MSE that is worse than any of the four (SN)(S)IS estimators. [12] also experimentally showed that the performance of $\hat{\beta}_{\mathrm{dr}}$ might be very bad in practice when the model is wrong.

We therefore define *intrinsic efficiency* as an additional desiderata, which prohibits this from occurring.

**Definition 2.5** (Intrinsic efficiency). *The asymptotic MSE of the estimator is smaller than that of any of $\hat{\beta}_{\text{sis}}, \hat{\beta}_{\text{is}}, \hat{\beta}_{\text{snsis}}, \hat{\beta}_{\text{snis}}, \hat{\beta}_{\text{dr}}$, irrespective of model specification.*

MDR can be seen as motivated by a variant of intrinsic efficiency against only DR (hence the # in Table 1). Although this is not precisely proven in [6], this arises as a corollary of our results. Nonetheless, MDR does not achieve full intrinsic efficiency against all above estimators.

# 3 REG and EMP for Contextual Bandit

None of the estimators above simultaneously satisfy all desired properties, Definitions 2.1–2.5. In the next sections, we develop new estimators that do. For clarity we first consider the simpler CB setting, where we write $(x, a, r)$ and $w$ instead of $(x_0, a_0, r_0)$ and $w_{0:0}$. We then start by showing how a modification to MDR ensures intrinsic efficiency. To obtain the other desiderata, we have to change how we choose the parameters. Regarding the intuitive detailed explanation, refer to Appendix A.

## 3.1 REG: Intrinsic Efficiency

When $T = 1$, $\hat{\beta}_d(m)$ in (2) becomes simply

$$\hat{\beta}_d(m) = \mathrm{E}_n\left[wr - \mathcal{F}(m)\right], \tag{3}$$

where $\mathcal{F}(m(x, a)) = wm(x, a) - \left\{\sum_{a \in A} m(x, a)\pi_e(a|x)\right\}$. By construction, $\mathrm{E}[\mathcal{F}(m)] = 0$ for every $m$. (M)DR, for example, use $m(x, a; \tau) = q(x, a; \tau)$.

Instead, we let

$$m(x, a; \zeta_1, \zeta_2, \tau) = \zeta_1 + \zeta_2 q(x, a; \tau),$$

for parameters $\tau$ and $\zeta = (\zeta_1, \zeta_2)$. This new choice has a special property: it includes both IS and DR estimators. Given any $\tau$, setting $\zeta_1 = 0, \zeta_2 = 0$ yields IS and setting $\zeta_1 = 0, \zeta_2 = 1$ gives (M)DR. This gives a simple recipe for intrinsic efficiency: estimate the variance of $\hat{\beta}_d(\zeta_1 + \zeta_2 q(x, a; \tau))$ and minimize it over $\tau, \zeta$. Because $\hat{\beta}_d(m)$ is unbiased, its variance is simply $\mathrm{E}\left[\{wr - \mathcal{F}(m)\}^2\right] - \beta^{*2}$. Therefore, over the parameter spaces $\Theta_\tau$ the (unknown) minimal variance choice is

$$(\zeta^*, \tau^*) = \underset{\zeta \in \mathbb{R}^2, \tau \in \Theta_\tau}{\arg\min} \ \mathrm{E}\left[\{wr - \mathcal{F}(\zeta_1 + \zeta_2 q(x, a; \tau))\}^2\right]. \tag{4}$$

We let the REG estimator be $\hat{\beta}_{\text{reg}} = \hat{\beta}_d(\hat{\zeta}_1 + \hat{\zeta}_2 q(x, a; \hat{\tau}))$ where we choose the parameters by minimizing the estimated variance:

$$(\hat{\zeta}, \hat{\tau}) = \underset{\zeta \in \mathbb{R}^2, \tau \in \Theta_\tau}{\arg\min} \ \mathrm{E}_n\left[\{wr - \mathcal{F}(\zeta_1 + \zeta_2 q(x, a; \tau))\}^2\right]. \tag{5}$$

To establish desired efficiencies, we prove the following theorem indicating that our choice of parameters does not inflate the variance. Note that it is not obvious because the plug-in some parameters generally causes an inflation of the variance.

**Theorem 3.1.** *When the optimal solution $(\zeta^*, \tau^*)$ in (4) is unique,*

$$\text{Asmse}[\hat{\beta}_{\text{reg}}] = n^{-1} \min_{\zeta \in \mathbb{R}^2, \tau \in \Theta_\tau} \mathrm{E}\left[\{wr - \mathcal{F}(\zeta_1 + \zeta_2 q(x, a; \tau))\}^2 - \beta^{*2}\right].$$

**Remark 3.1.** *For $\zeta = (\beta^*, 0)$, this asymptotic MSE is the same as the one of SNIS, $\text{var}[w(r - \beta^*)]$.*

From Theorem 3.1 we obtain the desired efficiencies. Importantly, to prove this, we note how the asymptotic MSEs of each of (SN)(S)IS and DR can be represented in the form $n^{-1}\mathrm{E}\left[\{wr - \mathcal{F}(\zeta_1 + \zeta_2 q(x, a; \tau))\}^2 - \beta^{*2}\right]$ for some $\zeta$ and $\tau$.

**Corollary 3.1.** *The estimator $\hat{\beta}_{\text{reg}}$ has local and intrinsic efficiency.*

**Remark 3.2** (Comparison to MDR). *REG is like MDR with an expanded model class. This class is carefully chosen to guarantee intrinsic efficiency. In addition, as another corollary, we have proven partial intrinsic efficiency for MDR against DR (just fix $\zeta = (0, 1)$ in (5)) where [6] only proved consistency of MDR. However, neither MDR nor REG satisfies boundedness and stability.*

**Remark 3.3** (SNREG). *Replacing weights $w$ by their self-normalized version $w/\mathrm{E}_n[w]$ in REG leads to SNREG. We explore this estimator in Appendix A and show it only gives 2-boundedness, does not give stability, and limits REG's intrinsic efficiency to be only against SN(S)IS and SNDR.*

## 3.2 EMP: Intrinsic Efficiency, Boundedness, and Stability

We next construct an estimator satisfying intrinsic efficiency as well as boundedness and stability. The key idea is to use empirical likelihood to choose the parameters [29–31]. Empirical likelihood is a nonparametric MLE commonly used in statistics [21]. We consider the control variate $m(x, a; \xi; \tau) = \xi + q(x, a; \tau)$ with parameters $\xi, \tau$ and $q(x, a; \tau) = t(x, a)^\top \tau$, where $t(x, a)$ is a $d_\tau$-dimensional vector of linear independent basis functions not including a constant. Then, an estimator for $\beta$ is defined as

$$\hat{\beta}_{\text{emp}} = \text{E}_n \left[ \hat{c}^{-1} \hat{\kappa}(x, a) \pi_e(a|x) r \right], \quad \text{where}$$

$$\hat{\kappa}(x, a) = \{\pi_b(a|x)[1 + \mathcal{F}(m(x, a; \hat{\xi}, \hat{\tau}))]\}^{-1}, \ \hat{c} = \text{E}_n \left[ \{1 + \mathcal{F}(m(x, a; \hat{\xi}, \hat{\tau}))\}^{-1} \right],$$

$$\hat{\xi}, \hat{\tau} = \underset{\xi \in \mathbb{R}, \tau \in \Theta_\tau}{\arg\max} \text{E}_n[\log\{1 + \mathcal{F}(m(x, a; \xi, \tau))\}]. \tag{6}$$

This is motivated by solving the dual problem of the following optimization problem formulated by the empirical likelihood:

$$\max_\kappa \sum_{i=1}^n \log \kappa^{(i)}, \ \text{s.t.} \sum_{i=1}^n \kappa^{(i)} \pi_b(a^{(i)}|x^{(i)}) = 1, \ \sum_{i=1}^n \kappa^{(i)} \pi_b(a^{(i)}|x^{(i)}) \mathcal{F}(m(x^{(i)}, a^{(i)}; \xi, \tau)) = 0.$$

The objective in an optimization problem (6) is a convex function; therefore, it is easy to solve. Then, the estimator $\hat{\beta}_{\text{emp}}$ has all the desirable finite-sample and asymptotic properties.

**Lemma 3.1.** *The estimator $\hat{\beta}_{\text{emp}}$ satisfies 1-boundedness and stability.*

**Theorem 3.2.** *The estimator $\hat{\beta}_{\text{emp}}$ has local and intrinsic efficiency, and*

$$\text{Asmse}[\hat{\beta}_{\text{emp}}] = n^{-1} \min_{\zeta \in \mathbb{R}, \tau \in \mathbb{R}^{d_\tau}} \text{E} \left[ \{wr - \mathcal{F}(\zeta + q(x, a; \tau))\}^2 - \beta^{*2} \right]. \tag{7}$$

Here, we have assumed the model is linear in $\tau$. Without this assumption, Theorem 3.2 may not hold. In the following section, we consider how to relax this assumption while maintaining local and intrinsic efficiency.

## 3.3 Practical REG and EMP

While REG and EMP have desirable theoretical properties, both have some practical issues. First, for REG, the optimization problem in (5) may be non-convex if $q(x, a; \tau)$ is not linear in $\tau$, as is the case in our experiment in Sec. 5.1 where we use a logistic model with 216 parameters. (The same issue exists for MDR.) Similarly, EMP estimator has the problem that there is no theoretical guarantee for intrinsic efficiency when $q(x, a; \tau)$ is not linear in $\tau$. Therefore, we suggest the following unified practical approach to selecting $\tau$ in a way that maintains the desired properties.

First, we estimate a parameter $\tau$ in $q(x, a; \tau)$ as in DM to obtain $\hat{\tau}$, which we assume has a limit, $\hat{\tau} \xrightarrow{p} \tau^\dagger$. Then, we consider solving the following optimization problems instead of (5) and (6) for REG and EMP, respectively

$$\hat{\zeta} = \underset{\zeta \in \mathbb{R}^2}{\arg\min} \text{E}_n \left[ \{wr - \mathcal{F}(m(x, a; \zeta, \hat{\tau}))\}^2 \right], \ \hat{\xi} = \underset{\xi \in \mathbb{R}^2}{\arg\max} \text{E}_n[\log\{1 + \mathcal{F}(m(x, a; \xi, \hat{\tau}))\}],$$

where $m(x, a; \zeta, \hat{\tau}) = \zeta_1 + \zeta_2 q(x, a; \hat{\tau})$ or $m(x, a; \xi, \hat{\tau}) = \xi_1 + \xi_2 q(x, a; \hat{\tau})$. This is a convex optimization problem with two dimensional parameters; thus, it is easy to solve.

Here, the asymptotic MSE of practical $\hat{\beta}_{\text{reg}}$ and $\hat{\beta}_{\text{emp}}$ are as follows.

**Theorem 3.3.** *The above plug-in-$\tau$ versions of $\hat{\beta}_{\text{reg}}$ and $\hat{\beta}_{\text{emp}}$ still satisfy local and intrinsic efficiency, and $\hat{\beta}_{\text{emp}}$ satisfies 1-boundedness and partial stability. Their asymptotic MSEs are*

$$n^{-1} \min_{\zeta \in \mathbb{R}^2} \text{E} \left[ \{wr - \mathcal{F}(\zeta_1 + \zeta_2 q(x, a; \tau^\dagger))\}^2 - \beta^{*2} \right]. \tag{8}$$

As a simple extension, we may consider multiple models for the Q-function. E.g, we can have two models $q_1(x, a; \tau_1)$ and $q_2(x, a; \tau_1)$ and let $m(x, a; \zeta, \hat{\tau}) = \zeta_1 + \zeta_2 q_1(x, a; \hat{\tau}_1) + \zeta_3 q_2(x, a; \hat{\tau}_2)$. Our results easily extend to provide intrinsic efficiency with respect to DR using any of these models.

## 4 REG and EMP for Reinforcement learning

We next present how REG and EMP extend to the RL setting. Some complications arise because of the multi-step horizon. For example, IS and SIS are different as opposed to the case $T = 1$.

### 4.1 REG for RL

We consider an extension of REG to a RL setting. First, we derive the variance of $\hat{\beta}_d(\{m_t\}_{t=0}^{T-1})$.

**Theorem 4.1.** *The variance of $\hat{\beta}_d(\{m_t\}_{t=0}^{T-1})$ is $n^{-1}\mathrm{E}[v(\{m_t\}_{t=0}^{T-1})]$, where $v(\{m_t\}_{t=0}^{T-1})$ is*

$$\sum_{t=0}^{T-1} \gamma^{2t}\omega_{0:t-1}^2 \mathrm{var}\left(\mathrm{E}[\sum_{k=t}^{T-1}\gamma^{k-t}\omega_{t:k}r_{k-t}|\mathcal{H}_t] - \left\{\omega_{t:t}m_t(x_t,a_t) - \sum_{a\in A} m_t(x_t,a)\pi_e(a|x_t)\right\}|\mathcal{H}_{t-1}\right). \tag{9}$$

To derive REG, we consider the class of estimators $\hat{\beta}_d(\{m_t\}_{t=0}^{T-1})$ where $m_t$ is $m_t(x_t,a_t;\zeta) = \zeta_{1t} + \zeta_{2t}q(x_t,a_t;\hat{\tau})$ for all $0 \le t \le T-1$. Then, we define an estimator $\hat{\zeta}$ and the optimal $\zeta^*$ as

$$\hat{\zeta} = \arg\min_{\zeta\in\mathbb{R}^2} \mathrm{E}_n[v(\{m_t(x_t,a_t;\zeta)\}_{t=0}^{T-1})], \quad \zeta^* = \arg\min_{\zeta\in\mathbb{R}^2} \mathrm{E}[v(\{m_t(x_t,a_t;\zeta)\}_{t=0}^{T-1})]. \tag{10}$$

REG is then defined as $\hat{\beta}_{\mathrm{reg}}^{T-1} = \hat{\beta}_d(\{\hat{\zeta}_{1t} + \hat{\zeta}_{2t}q(x,a;\hat{\tau})\}_{t=0}^{T-1})$, where following our discussion in Section 3.3, $\hat{\tau}$ is given by fitting as in DM/DR. Theoretically, we could also choose $\tau$ to minimize eq. (9), but that can be computationally intractable.

A similar argument to that in Section 3.1 shows that a data-driven parameter choice induces no inflation in asymptotic MSE. Therefore, the asymptotic MSE of the estimator $\hat{\beta}_{\mathrm{reg}}$ is minimized among the class of estimators $\hat{\beta}_d(\{\zeta_{1t} + \zeta_{2t}q(x_t,a_t;\hat{\tau})\}_{t=0}^{T-1}))$. This implies that the asymptotic MSE of $\hat{\beta}_{\mathrm{reg}}$ is smaller than $\hat{\beta}_{\mathrm{sis}}$ and $\hat{\beta}_{\mathrm{dr}}$ because $\hat{\beta}_{\mathrm{sis}}$ corresponds to the case $\zeta_t = (0,0)$ and $\hat{\beta}_{\mathrm{dr}}$ corresponds to the case $\zeta_t = (0,1)$. In addition, we can prove that the estimator $\hat{\beta}_{\mathrm{reg}}^{T-1}$ is more efficient than $\hat{\beta}_{\mathrm{snsis}}$. To prove this, we introduce the following lemma.

**Lemma 4.1.**

$$\mathrm{Asmse}[\hat{\beta}_{\mathrm{snis}}] = n^{-1}\sum_{t=0}^{T-1}\mathrm{E}\left[\gamma^{2t}\omega_{0:t-1}^2\mathrm{var}\left(\omega_{t:t}\left(\mathrm{E}\left[\sum_{k=t}^{T-1}\gamma^{k-t}\omega_{t+1:k}r_{k-t}|\mathcal{H}_t\right] - \beta_t^*\right)|\mathcal{H}_{t-1}\right)\right],$$

*where $\beta_t^* = \mathrm{E}[\omega_{0:t}r_t]$.*

We note that setting $\zeta_t = (\beta_t^*, 0)$ in eq. (9) recovers the above. This suggests the following theorem.

**Theorem 4.2.** *The estimator $\hat{\beta}_{\mathrm{reg}}^{T-1}$ is locally and intrinsically efficient.*

**Remark 4.1.** *Practically, when the horizon is long, there may be too many parameters to optimize, which can causes overfitting. That is, although there is no inflation in MSE asymptotically, there may be issues in finite samples. To avoid this problem, some constraint or regularization should be imposed on the parameters. Here we will consider the estimator $\hat{\beta}_{\mathrm{reg}}^k$ ($0 \le k \le T-1$) given by $\hat{\beta}_d(\{m_t(x_t,a_t;\hat{\zeta})\}_{t=0}^{T-1})$ for the constrained control variates:*

$$m_t(x_t,a_t;\zeta) = \begin{cases} \zeta_{t1} + \zeta_{t2}q(x_t,a_t;\hat{\tau}) \, (0 \le t < k), \\ \zeta_{k1} + \zeta_{k2}q(x_t,a_t;\hat{\tau}) \, (k \le t \le T-1). \end{cases}$$

*The estimator $\hat{\beta}_{\mathrm{reg}}^{T-1}$ corresponds to the originally introduced estimator. We can also obtain theoretical guarantees of $\hat{\beta}_{\mathrm{reg}}^k$ for $k \ne T-1$. For details, see Appendix C.*

### 4.2 EMP for RL

First, we define a control variate:

$$g(\mathcal{D}_{x,a};\xi,\hat{\tau}) = \sum_{t=0}^{T-1}\gamma^t\left(\omega_{0:t}m_t(x_t,a_t;\xi,\hat{\tau}) - \omega_{0:t-1}\left\{\sum_{a\in A}m_t(x_t,a;\xi,\hat{\tau})\pi_e(a|x_t)\right\}\right).$$

Table 2: SatImage (RMSE $\times 1000$ )

| Behavior policy | DM1 | DM2 | IS | SNIS | DR | MDR | REG | EMP |
|---|---|---|---|---|---|---|---|---|
| $0.7\pi_d + 0.3\pi_u$ | 18.1 | 12.2 | 6.7 | 4.0 | 3.0 | 3.8 | **2.8** | **2.8** |
| $0.4\pi_d + 0.6\pi_u$ | 49.2 | 30.5 | 12.0 | 5.6 | 5.0 | 5.3 | **4.4** | **4.4** |
| $0.0\pi_d + 1.0\pi_u$ | 128.6 | 71.7 | 26.0 | **12.7** | 18.0 | 14.4 | **13.6** | 13.7 |

Table 3: Pageblock (RMSE $\times 1000$ )

| Behavior policy | DM1 | DM2 | IS | SNIS | DR | MDR | REG | EMP |
|---|---|---|---|---|---|---|---|---|
| $0.7\pi_d + 0.3\pi_u$ | 21.8 | 2.6 | 8.5 | 3.4 | **1.4** | 2.3 | 1.5 | **1.4** |
| $0.4\pi_d + 0.6\pi_u$ | 32.4 | 5.6 | 13.4 | 4.0 | 2.7 | 3.4 | **2.5** | **2.4** |
| $0.0\pi_d + 1.0\pi_u$ | 62.0 | 16.0 | 27.2 | 6.5 | 7.2 | 6.4 | **4.9** | **4.9** |

Table 4: PenDigits (RMSE $\times 1000$ )

| Behavior policy | DM1 | DM2 | IS | SNIS | DR | MDR | REG | EMP |
|---|---|---|---|---|---|---|---|---|
| $0.7\pi_d + 0.3\pi_u$ | 8.1 | 8.2 | 6.1 | 2.8 | 1.5 | 2.2 | **1.4** | **1.4** |
| $0.4\pi_d + 0.6\pi_u$ | 19.4 | 17.4 | 10.7 | 3.9 | 2.2 | 3.4 | **2.1** | **2.0** |
| $0.0\pi_d + 1.0\pi_u$ | 58.6 | 56.0 | 29.6 | 9.9 | 11.1 | **9.4** | **9.4** | 9.5 |

By setting $m_t(x_t, a_t; \xi, \hat{\tau}) = \xi_{1t} + \xi_{2t} q(x_t, a_t; \hat{\tau})$, define $\hat{\xi}$;

$$\hat{\xi}(\hat{\tau}) = \arg\max_{\xi \in \mathbb{R}^2} \mathrm{E}_n[\log\{1 + g(\mathcal{D}_{x,a}; \xi, \hat{\tau})\}].$$

Then, an estimator $\hat{\beta}_{\mathrm{emp}}^{T-1}$ is defined as

$$\hat{\beta}_{\mathrm{emp}}^{T-1} = \mathrm{E}_n\left[\sum_{t=0}^{T-1} \omega_{0:t}\gamma^t r_t \frac{\hat{c}^{-1}}{1 + g(\mathcal{D}_{x,a}; \hat{\xi}, \hat{\tau})}\right], \ \hat{c} = \mathrm{E}_n\left[\frac{1}{1 + g(\mathcal{D}_{x,a}; \hat{\xi}, \hat{\tau})}\right].$$

This estimator has the same efficiencies as $\hat{\beta}_{\mathrm{reg}}^{T-1}$ because the asymptotic MSE is the same. Importantly, the estimator $\hat{\beta}_{\mathrm{emp}}^{T-1}$ also satisfies a 1-boundedness and stability.

**Theorem 4.3.** *The asymptotic MSE of the estimator $\hat{\beta}_{\mathrm{emp}}^{T-1}$ is the same as that of $\hat{\beta}_{\mathrm{reg}}^{T-1}$. Hence, it is also locally and intrinsically efficient. It also satisfies 1-boundeness and stability.*

## 5 Experiments

### 5.1 Contextual Bandit

We evaluate the OPE algorithms using the standard classification data-sets from the UCI repository. Here, we follow the same procedure of transforming a classification data-set into a contextual bandit data set as in [5, 6]. Additional details of the experimental setup are given in Appendix E.

We first split the data into training and evaluation. We make a deterministic policy $\pi_d$ by training a logistic regression classifier on the training data set. Then, we construct evaluation and behavior policies as mixtures of $\pi_d$ and the uniform random policy $\pi_u$. The evaluation policy $\pi_e$ is fixed at $0.9\pi_d + 0.1\pi_u$. Three different behavior policies are investigated by changing a mixture parameter.

Here, we compare the (practical) REG and EMP with DM, SIS, SNIS, DR, and MDR on the evaluation data set. First, two Q-functions $\hat{q}_1(x, a), \hat{q}_2(x, a)$ are constructed by fitting a logistic regression in two ways with a $l1$ or $l2$ regularization term. We refer them as DM1 and DM2. Then, in DR, we use a mixture of Q-functions $0.5\hat{q}_1 + 0.5\hat{q}_2$ as $m(x, a)$. For MDR, we use a logistic function as $m(x, a)$ and we use SGD to solve the resulting non-convex high-dimensional optimization (e.g., for SatImage we have $6(\text{number of actions}) \times 36(\text{number of covariates})$ parameters). We use $m(x, a; \zeta) = \zeta^\top(1, \hat{q}_1, \hat{q}_2)$ in REG and $m(x, a; \xi) = \xi^\top(1, \hat{q}_1, \hat{q}_2)$ in EMP.

The resulting estimation RMSEs (root mean square error) over 200 replications of each experiment are given in Tables 2–4, where we highlight in bold the best two methods in each case. We first find that REG and EMP generally have overall the best performance. Second we see that this arises

Table 5: Windy GridWorld (RMSE)

| Size | DM | SIS | SNSIS | DR | MDR | REG | EMP |
|------|-----|------|-------|------|------|------|------|
| 250  | 2.9 | 0.64 | 0.49  | 0.17 | 0.28 | **0.09** | **0.09** |
| 500  | 2.8 | 0.53 | 0.34  | 0.11 | 0.21 | **0.06** | **0.06** |
| 750  | 2.6 | 0.39 | 0.29  | 0.09 | 0.14 | **0.05** | **0.05** |

Table 6: Cliff Walking (RMSE)

| Size | DM | SIS | SNSIS | DR | MDR | REG | EMP |
|------|-----|-----|-------|-----|-----|------|------|
| 1000 | 7.7 | 3.6 | 2.9   | 2.5 | 2.3 | **2.1** | **2.1** |
| 2000 | 6.0 | 3.2 | 2.4   | 2.3 | 2.2 | **1.6** | **1.5** |
| 3000 | 6.8 | 3.1 | 2.2   | 2.2 | 2.0 | **1.2** | **1.1** |

Table 7: Mountain Car (RMSE)

| Size | DM | SIS | SNSIS | DR | MDR | REG | EMP |
|------|------|-----|-------|-----|-----|------|------|
| 1000 | 9.8  | 4.2 | 3.7   | 1.9 | 1.9 | **1.7** | **1.7** |
| 2000 | 10.6 | 3.3 | 2.9   | 1.6 | 1.6 | **1.2** | **1.2** |
| 3000 | 8.2  | 2.4 | 1.8   | 1.4 | 1.5 | **1.0** | **1.0** |

because they achieve similar RMSE to SNIS when SNIS performs well and similar RMSE to (M)DR when (M)DR performs well, which is thanks to the intrinsic efficiency property. Whereas REG's and EMP's intrinsic efficiency is visible, MDR still often does slightly worse than DR despites its partial intrinsic efficiency, which can be attributed to optimizing too many parameters leading to overfitting in the sample size studied.

## 5.2  Reinforcement Learning

We next compare the OPE algorithms in three standard RL setting from OpenAI Gym [3]: Windy GridWorld, Cliff Walking, and Mountain Car. For further detail on each see Appendix E. We again split the data into training and evaluation. In each setting we consider varying evaluation dataset sizes. In each setting, a policy $\pi_d$ is computed as the optimal policy based on the training data using Q-learning. The evaluation policy $\pi_e$ is then set to be $(1-\alpha)\pi_d + \alpha\pi_u$, where $\alpha = 0.1$. The behavior policy is defined similarly with $\alpha = 0.2$ for Windy GridWorld and Cliff Walking and with $\alpha = 0.15$ for Mountain Car. We set the discounting factor to be $1.0$ as in [6].

We compare the (practical) REG, EMP with $k = 2$ with DM, SIS, SNSIS, DR, MDR on the evaluation data set generated by a behavior policy. A Q-function model is constructed using an off-policy TD learning [27]. This is used in DM, DR, REG, and EMP. For MDR, we use a linear function for $m(x, a)$ in order to enable tractable optimization given the many parameters due to long horizons.

We report the resulting estimation RMSEs over 200 replications of each experiment in Tables 5–7. We find that the modest benefits we gained in one time step in the CB setting translate to significant outright benefits in the longer horizon RL setting. REG and EMP consistently outperform other methods. Their RMSEs are indistinguishable except for one setting where EMP has slightly better RMSE. These results highlight how the theoretical properties of intrinsic efficiency, stability, and boundedness can translate to improved performance in practice.

## 6  Conclusion and Discussion

We studied various desirable properties for OPE in CB and RL. Finding that no existing estimator satisfies all of them, we proposed two new estimators, REG and EMP, that satisfy consistency, local efficiency, intrinsic efficiency, 1-boundedness, and stability. These theoretical properties also translated to improved comparative performance in a variety of CB and RL experiments.

In practice, there may be additional modifications that can further improve these estimators. For example, [32, 35] propose hybrid estimators that blend or switch to DM when importance weights are very large. This reportedly works very well in practice but may make the estimator inconsistent under misspecification unless blending vanishes with $n$. In this paper, we focused on consistent estimators. Also these do not satisfy intrinsic efficiency, 1-boudedness, or stability. Achieving these properties with blending estimators remains an important next step.

## Acknowledgements

This material is based upon work supported by the National Science Foundation under Grant No. 1846210.

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
