[Supplementary Material]

Table 8: Summary of notations

| | |
|---|---|
| $\pi_e(a\|x)$ | Target policy |
| $\pi_b(a\|x)$ | Exploration policy |
| $\beta^*$ | Parameter of interest $\beta_T^{\pi_e}$ |
| $\mathbb{P}, \mathrm{E}[\cdot]$ | Expectation with respect to a behavior policy |
| $\mathrm{var}[\cdot]$ | Variance |
| $\mathrm{Asmse}[\cdot]$ | Asymptotic variance |
| $\mathbb{P}_n, \mathrm{E}_n$ | Empirical approximation based on a set of samples from a behavior policy |
| $\mathbb{G}_n$ | Empirical process $\sqrt{n}(\mathbb{P}_n - \mathbb{P})$ |
| $q(x, a; \tau)$ | Model for Q-function with parameter $\tau$ |
| $\omega_{t_1:t_2}$ | Cumulative importance ratio $\prod_{t=t_1}^{t_2} \pi_e(a_t\|x_t)/\pi_b(a_t\|x_t)$ |
| $\zeta$ | Parameter in $m(x)$ for REG, SNREG |
| $\xi$ | Parameter in $m(x)$ for EMP |
| $R_{\max}$ | An upper bound of the reward function |
| $\mathcal{H}_{T-1}$ | $(x_0, a_0, r_0, \cdots, x_{T-1}, a_{T-1}, r_{T-1})$ in T-step trajectory |
| $x^{(i)}$ | $i$-th sample |
| $\xrightarrow{p}$ | Convergence in probability |

# A   Additional Intuitive explanation

Here, we add several intuitive explanations, which can not be included due to the space limit.

**Section 2**

We explain Figure 1. The figure seeks to illustrate the local and intrinsic efficiency uniquely achieved by our new estimators. It shows the ordering of asymptotic MSEs: if Q-functions are well specified, both our proposed estimators (EMP, REG) and DR achieve the same efficiency bound, which IS and SNIS do not; if Q-functions are misspecified, the efficiency bound is not achieved, yet our proposed estimators will still have better MSE than DR, IS, and SNIS. Note that we use the terminology of the well-specification and misspecification for parametric models.

**Section 3**

In section 3,1, the core idea is adding parameters $\zeta_1$ and $\zeta_2$ in addition to $\tau$. These parameters ensures the intrinsic efficiency since the existence of $\zeta_1$ gives the superiority over SNIS, and the existence of $\zeta_2$ gives the superiority over IS.

In section 3.2, the key idea is using an empirical likelihood [22]. Empirical likelihood is known as an optimal way to utilize some constraints (moment conditions). As it is known as nonparametric MLE, the formulation is seen as maximizing each empirical weight following some imposed constraints. Since it is hard to directly solve the original problem, people usually solve the dual formulation. For the current context, control variate is seen as a constraint; so we add the following constraint

$$\sum_{i=1}^{n} \kappa^{(i)} \pi_b(a^{(i)}|x^{(i)}) \mathcal{F}(m(x^{(i)}, a^{(i)}; \xi, \tau)) = 0.$$

The another constraint

$$\sum_{i=1}^{n} \kappa^{(i)} \pi_b(a^{(i)}|x^{(i)}) = 1$$

is imposed so that each weight corresponds to the probability weight. Under these constraints, the log of the product of weights (nonparametric log–likelihood)

$$\log \left( \prod_{i=1}^{n} \kappa^{(i)} \right)$$

is optimized.

The critical idea in Section 3.3 is reducing the first stage optimization in Section 3.1 and 3.2 to the second stage optimization. More specifically, $\tau$ is optimized in the first stage; then, $\zeta$ is optimized. The second stage optimization has a computational advantage over the first stage optimization. An

intrinsic property is still retained since $\zeta$ governs the intrinsic efficiency property rather than $\tau$. In addition, the local efficiency is sill retained since $\tau$ is optimized in the first stage.

## B SNREG (self-normalized REG)

Herein, we construct an estimator exhibiting partial intrinsic efficiency, 2-boundedness and partial stability based on a self-normalized estimator [24, 32]. The partial intrinsic efficiency means that the resulting estimator's asymptotic MSE is smaller than SNDR and SNIS. Further, partial stability is defined as follows.

**Definition B.1** (Partial stability)**.** *An estimator satisfies the stability when $\hat{\tau}$ does not depend on the reward.*

This condition indicates that the variance can be still bounded after defining the ratio and the estimated Q-function. The DM, SNDR have been easily proved to have this property. In addition, in the following proof section, we prove that the practical EMP also possesses this property.

Consider a family of unbiased estimators: $\hat{\beta}_{\mathrm{snd}}(m)$ as a solution to

$$\mathrm{E}_n\left[\beta - \left\{\sum_{a\in A} m(x,a)\pi_e(a|x)\right\} - \frac{\omega(a,x)}{\mathrm{E}_n[\omega(a,x)]}\{r - m(x,a)\}\right] = 0,$$

where $\pi_e(a|x)/\pi_b(a|x) = \omega(a,x)$. The SNDR estimator is subsequently defined as $\hat{\beta}_{\mathrm{sndr}} = \hat{\beta}_{\mathrm{snd}}(q(x,a;\hat{\tau}))$. First, the range of this estimator is $[0, 2R_{\max}]$. Therefore, tihs satisfies 2-boundedness and partial stability. In addition, this satisfies the consistency for an arbitrary choice of $m(x,a)$. By selecting $\zeta_1 + \zeta_2 q(a,x;\tau)$ as $m(x,a)$, this class is also observed to include a SNIS estimator setting $\zeta = (1,0)$, and a SNDR estimator setting $\zeta = (0,1)$. However, this class does not include an IS estimator.

The asymptotic MSE is calculated as follows.

**Theorem B.1.** *The term* $\mathrm{Asmse}[\hat{\beta}_{\mathrm{snd}}]$ *is* $n^{-1}V_{\mathrm{snd}}(m)$, *where* $V_{\mathrm{snd}}(m)$ *is*

$$\mathrm{var}\left[\omega(a,x)\left(r - m(x,a)\right) - \left\{\sum_{a\in A} m(x,a)\pi_e(a|x)\right\}\right]$$

$$+ \mathrm{E}\left[\omega(a,x)(r - m(x,a))\right]^2 \mathrm{var}\left[\omega(a,x)\right]$$

$$- 2\left(\mathrm{E}\left[w(a,x)^2(r - m(x,a)) - \omega(a,x)\sum_{a\in A}\pi_e(a|x)m(a,x)\right] - \beta^*\right)\mathrm{E}\left[\omega(a,x)(r - m(x,a))\right].$$

By minimizing the empirical approximation of the aforementioned asymptotic MSE with respect to $\zeta_1, \zeta_2$ and $\tau$ and plugging-in as

$$(\hat{\zeta}, \hat{\tau}) = \operatorname*{arg\,min}_{\zeta\in\mathbb{R}^2, \tau\in\Theta_\tau} \hat{V}_{\mathrm{snd}}(m(x,a;\zeta,\tau)),$$

we obtain the estimator $\hat{\beta}_{\mathrm{snreg}} = \hat{\beta}_{\mathrm{snd}}(m(x,a;\hat{\zeta},\hat{\tau}))$. Here, $(\hat{\zeta}, \hat{\tau})$ converges in probability to $(\zeta^*, \tau^*)$

$$(\zeta^*, \tau^*) = \operatorname*{arg\,min}_{\zeta\in\mathbb{R}^2, \tau\in\Theta_\tau} V_{\mathrm{snd}}(m(x,a;\zeta,\tau)). \tag{11}$$

The asymptotic MSE of $\hat{\beta}_{\mathrm{snreg}}$ is given as follows.

**Theorem B.2.** *Under the assumption that the optimization problem in* (11) *has a unique solution,*

$$\mathrm{Asmse}[\hat{\beta}_{\mathrm{snreg}}] = n^{-1} \min_{\zeta\in\mathbb{R}^2, \tau\in\Theta_\tau} V_{\mathrm{snd}}(m(x,a;\zeta,\tau)).$$

*The asymptotic MSE is smaller than those of the SNIS and SNDR.*

**Theorem B.3.** *The estimator* $\hat{\beta}_{\mathrm{snreg}}$ *is locally efficient.*

Table 9: SatImage ($\times 1000$)

| Behavior policy | DR | SNDR | MDR | REG | SNREG | EMP |
|---|---|---|---|---|---|---|
| $0.7\pi_d + 0.3\pi_u$ | 3.0 | 3.0 | 3.8 | 2.8 | 2.8 | 2.8 |
| $0.4\pi_d + 0.6\pi_u$ | 5.0 | 5.0 | 5.3 | 4.4 | 4.4 | 4.4 |
| $0.0\pi_d + 1.0\pi_u$ | 18.0 | 17.8 | 14.4 | 13.6 | 13.6 | 13.7 |

Table 10: Pageblock ($\times 1000$)

| Behavior policy | DR | SNDR | MDR | REG | SNREG | EMP |
|---|---|---|---|---|---|---|
| $0.7\pi_d + 0.3\pi_u$ | 1.4 | 1.4 | 2.3 | 1.5 | 1.4 | 1.4 |
| $0.4\pi_d + 0.6\pi_u$ | 2.7 | 2.6 | 3.4 | 2.5 | 2.5 | 2.4 |
| $0.0\pi_d + 1.0\pi_u$ | 7.2 | 7.3 | 6.4 | 4.9 | 4.9 | 4.9 |

Table 11: PenDigits($\times 1000$)

| Behavior policy | DR | SNDR | MDR | REG | SNREG | EMP |
|---|---|---|---|---|---|---|
| $0.7\pi_d + 0.3\pi_u$ | 1.5 | 1.5 | 2.2 | 1.4 | 1.4 | 1.4 |
| $0.4\pi_d + 0.6\pi_u$ | 2.2 | 2.2 | 3.4 | 2.1 | 2.1 | 2.0 |
| $0.0\pi_d + 1.0\pi_u$ | 11.1 | 10.8 | 9.4 | 9.4 | 9.4 | 9.5 |

*Proof.* The variance reaches an efficiency bound: $\zeta_1 = 0, \zeta_2 = 1$ and $\tau = \tau^*$, noting

$$\mathrm{E}[w(a,x)\{r - m(x,a)\}] = 0.$$

$\square$

Table 9-11 shows the experimental result of SNREG. The performance of SNREG is quite similar to those of REG, SNREG and EMP.

## C  Theoretical property of $\hat{\beta}^0_{\mathrm{reg}}$

Herein, we provide some theoretical property of $\hat{\beta}^0_{\mathrm{reg}}$. In fact, the variance of $\hat{\beta}^0_{\mathrm{reg}}$ is smaller than the following estimator:

$$\hat{\beta}_{\mathrm{sn2sis}} = \mathrm{E}_n \left[ \sum_{t=0}^{T-1} \frac{\omega_{0:t}}{\mathrm{E}_n[\omega_{0:T-1}]} \gamma^t r_t \right].$$

The difference between this estimator and $\hat{\beta}_{\mathrm{snsis}}$ is that the denominator is $\mathrm{E}_n[\omega_{0:T-1}]$ instead of $\mathrm{E}_n[\omega_{0:t-1}]$.

**Theorem C.1.** *The asymptotic MSE of $\hat{\beta}^0_{\mathrm{reg}}$ is smaller than those of $\hat{\beta}_{\mathrm{sis}}$, $\hat{\beta}_{\mathrm{sn2is}}$ and $\hat{\beta}_{\mathrm{dr}}$.*

## D  Proofs

The assumption is as follows.

**Assumption D.1.** *(a1) Parameter space $\Theta_\tau$ is compact and sufficiently large, (a2) the term $|q(x, a; \tau)| \leq R_{\max}$, (a3) the optimal solution $(\zeta^*, \tau^*)$ in (4) is unique.*

Note that we have assumed (a1) and (a2) for all of theorems. Regarding (a3), we have assumed for Theorem 3.1. In addition, we have assumed that the reward $r$ and the cumulative ratio $w_{t_1:t_2}$ are bounded in the main paper. These condition (uniform boundedness of reward and cumulative ratio) can be relaxed to each theorem when discussing asymptotic properties. However, for simplicity, we assumed these conditions.

*Proof of Theorem 3.1.* We denote $m^* = \zeta_1^* + \zeta_2^* q(x, a; \tau^*)$, $\hat{m} = \hat{\zeta}_1 + \hat{\zeta}_2 q(x, a; \hat{\tau})$ and $u(m)$ as

$$\omega(a,x)r + \left\{ \sum_{a \in A} m(x,a)\pi_e(a|x) \right\} - \omega(a,x)m(x,a).$$

We prove two lemmas first.

**Lemma D.1.** $\hat{\zeta} \xrightarrow{p} \zeta^*$ and $\hat{\tau} \xrightarrow{p} \tau^*$.

*Proof.* First, we define a space $\Theta_\zeta$, which always includes $\hat{\zeta}$. We can take a compact set as $\Theta_\zeta$ noting that is uniquely defined fixing $\tau$, $\hat{\zeta}$ and the all of assumptions.

Then, based on Lemma 2.4 in [20], an uniform convergence condition:

$$\sup_{\tau \in \Theta_\tau, \zeta \in \Theta_\tau} |(\mathbb{P}_n - \mathbb{P})\{u(\zeta_1 + \zeta_2 q(x, a; \tau))\}^2| \xrightarrow{p} 0$$

is satisfied using an assumption (a1) and the fact from (a2) that $u(\zeta_1 + \zeta_2 q(x, a; \tau))^2$ is bounded uniformly over $\zeta \in \mathbb{R}^2$ and $\tau \in \Theta_\tau$.

Then, by using Theorem 5.7 in [34], the statement holds from (a1), (a3) and the above uniform convergence condition. □

**Lemma D.2.** $\mathbb{G}_n[u(\hat{m})] - \mathbb{G}_n[u(m^*)] = o_p(1)$.

*Proof.* Based on Lemma 19.24 in [34], we have to confirm two statements; (1): for some $\delta > 0$, the class $\{u(\zeta_1 + \zeta_2 q(x, a; \tau)); |\zeta - \zeta^*| < \delta, |\tau - \tau^*| < \delta\}$ is a Donsker class, (2) the term $\mathrm{E}[(u(\hat{m}) - u(m^*))^2]$ converges in probability to 0.

The first condition is satisfied using the assumption (a1) and the fact from (a2) that $u(\zeta_1 + \zeta_2 q(x, a; \tau))$ is bounded uniformly over $\zeta \in \mathbb{R}^2$ and $\tau \in \Theta_\tau$, based on Example 19.7 in [34].

The second condition is satisfied as follows. First, $\hat{m}$ converges in probability to $m^*$ from Lemma D.1 by continuous mapping theorem. In addition, $\{u(\zeta_1 + \zeta_2 q(x, a; \tau)); \zeta \in \mathbb{R}^2, \tau \in \Theta_\tau\}$ is uniformly integrable from the assumption (a2). Then, it is verified by Lebesgue convergence theorem. □

We go back to the main proof. Here, we want to know the behavior of $\sqrt{n}(u(\hat{m}) - \beta^*)$. This is decomposed as

$$\begin{aligned}
\sqrt{n}(u(\hat{m}) - \beta^*) &= \mathbb{G}_n[u(\hat{m})] - \mathbb{G}_n[u(m^*)] \\
&\quad + \mathbb{G}_n[u(m^*)] \\
&\quad + \sqrt{n}(\mathrm{E}[u(\hat{m})] - \beta^*).
\end{aligned}$$

The first term is $o_p(1)$ by Lemma D.2. The third term $\sqrt{n}(\mathrm{E}[u(\hat{m})] - \beta^*)$ is 0 from the construction. Then, it is found that the influence function of the estimator is $u(m^*)$, that is,

$$\sqrt{n}(u(\hat{m}) - \beta^*) = \mathbb{G}_n[u(m^*)] + o_p(1).$$

Thus, the asymptotic MSE of $\hat{\beta}_d(\hat{\zeta}_1 + \hat{\zeta}_2 q(x, a; \hat{\tau}))$ is the same as the variance of $\hat{\beta}_d(\zeta_1^* + \zeta_2^* q(x, a; \tau^*))$. This concludes the proof. □

*Proof of Corollary 3.1.* We prove each statement as follows.

**Local efficiency** By setting $\zeta = (0, 1), \tau = \tau^*$ in Theorem 3.1, it achieves the efficiency bound.

It is obvious because the asymptotic variance of $\hat{\beta}_{\mathrm{reg}}$ estimator is represented as

$$n^{-1} \underset{\zeta \in \mathbb{R}^2, \tau \in \Theta_\tau}{\arg\min} \mathrm{E}[\{wr - \mathcal{F}(\zeta_1 + \zeta_2 q(x; \tau))\}^2].$$

**Intrinsic efficiency** We notice that the asymptotic variance of each estimator is represented as $n^{-1}\mathrm{E}[\{wr - \mathcal{F}(\zeta_1 + \zeta_2 q(x; \tau))\}^2]$. The SIS estimator corresponds to the case $\zeta = (0, 0)$. The SNSIS estimator corresponds to the case $\zeta = (\beta^*, 0)$. The DR estimator corresponds to the case $\zeta = (0, 1)$ and $\tau = \tau^\dagger$, where $\tau^\dagger$ is some convergence point of $\hat{\tau}$. □

*Proof of Lemma 3.1.* Because of the first order condition in (6), the following equation holds:

$$\sum_{i=1}^n \hat{\kappa}^{(i)} \pi_b(a^{(i)}|x^{(i)})(w(x^{(i)}, a^{(i)}) - 1) = 0,$$

where $\hat{\kappa}^{(i)} = \hat{\kappa}(a^{(i)}|x^{(i)}; \hat{\xi}, \hat{\tau})$. Then,

$$\sum_{i=1}^{n} \hat{\kappa}^{(i)}(\pi_e(a^{(i)}|x^{(i)}) - \pi_b(a^{(i)}|x^{(i)})) = 0$$

Regarding the 1-boundedness, it is proved as follows.

$$\hat{\beta}_{\text{emp}} = \frac{1}{n}\sum_{i=1}^{n} \hat{c}(\mathcal{D}_{x,a}; \hat{\xi}, \hat{\tau})^{-1}\hat{\kappa}(\mathcal{D}_{x,a}; \hat{\xi}, \hat{\tau})\pi_e(a^{(i)}|x^{(i)})r^{(i)}$$

$$\leq \frac{1}{n}\sum_{i=1}^{n} \hat{c}(\mathcal{D}_{x,a}; \hat{\xi}, \hat{\tau})^{-1}\hat{\kappa}(\mathcal{D}_{x,a}; \hat{\xi}, \hat{\tau})\pi_e(a^{(i)}|x^{(i)})R_{\max}$$

$$= R_{\max}.$$

From the third line to the fourth line, we use a definition of $\hat{c}$.

Regarding the partial stability, noting $\hat{\xi}$ and $\hat{\tau}$ are a function of $\mathcal{D}_{x,a}$ base on the form of optimization problem (6), it is proved as follows;

$$\text{var}[\hat{\beta}_{\text{emp}}|\mathcal{D}_{x,a}] = \frac{1}{n}\sum_{i=1}^{n} \left\{ \hat{c}(\mathcal{D}_{x,a}; \hat{\xi}, \hat{\tau})^{-1}\hat{\kappa}(\mathcal{D}_{x,a}; \hat{\xi}, \hat{\tau})\pi_e(a^{(i)}|x^{(i)}) \right\}^2 \text{var}[r^{(i)}|\mathcal{D}_{x,a}]$$

$$\leq \frac{1}{n}\sum_{i=1}^{n} \left\{ \hat{c}(\mathcal{D}_{x,a}; \hat{\xi}, \hat{\tau})^{-1}\hat{\kappa}(\mathcal{D}_{x,a}; \hat{\xi}, \hat{\tau})\pi_e(a^{(i)}|x^{(i)}) \right\}^2 \sigma^2$$

$$\leq \sigma^2.$$

From the second line to the third line, we have used the fact that $\max_b \sum b_i^2$ such that $\sum b_i = 1$ is 1. $\qquad \square$

*Proof of Theorem 3.2.* First, we prove $\hat{\xi} \xrightarrow{P} 0$ and $\hat{\tau} \xrightarrow{P} 0$. Define $(\xi, \tau^\top)^\top = \psi$.

**Lemma D.3.** $\hat{\psi} \xrightarrow{P} 0$

*Proof.* We use Theorem 5.7 in [34]. Here, note that

$$\mathcal{F}(\xi + \tau^\top t(x, a)) = \psi^\top g(x, a),$$

where $g(x, a) = (\mathcal{F}(1), \mathcal{F}(t(x, a)))^\top$ and the estimator $\hat{\psi}$ is an M-estimator defined by maximizing:

$$E_n[\log(1 + \psi^\top g(x, a))].$$

The uniform convergence condition is proved similarly as the proof in Theorem 3.1 based on (a1) and (a2). What we have to show is $E[\log(1 + \psi^\top g(x, a))]$ takes a maximum over $\psi \in \mathbb{R}^{d_\psi}$ if and only if $\psi = 0$. This comes from the Jensen inequality:

$$E[\log(1 + \psi^\top g(x, a))] \leq \log E[(1 + \psi^\top g(x, a))]$$
$$= \log\{1 + \psi^\top E[g(x, a)]\} = 0,$$

and a corresponding Hessian is a negative definite matrix. $\qquad \square$

Then, we can state that $\hat{c}$ also converges in probability to 1

**Lemma D.4.** $\hat{c} \xrightarrow{P} 1$.

*Proof.* We have

$$|\hat{c} - 1| \leq |(\mathbb{P}_n - \mathbb{P})\{1 + \mathcal{F}(m(x, a; \hat{\psi}))\}^{-1}| + |\mathbb{P}[\{1 + \mathcal{F}(m(x, a; \hat{\psi}))\}^{-1}] - 1|.$$

The first term converges in probability to 0 from the uniform convergence property based on the assumption (a1) and (a2). The second term also converges in probability to 0 from the continuous mapping theorem, noting $\hat{\psi} \xrightarrow{P} 0$. $\qquad \square$

Then, we show the following lemma.

**Lemma D.5.**

$$\sqrt{n}\left(\mathbb{P}_n\left(\frac{\pi_e r}{\pi_b{}'(\hat{c},\hat{\psi})}\right)-\beta^*\right)=\sqrt{n}\left(\mathbb{P}_n\left(\frac{\pi_e r}{\pi_b}-\psi^* g(x,a)\right)-\beta^*\right)+o_p(1),$$

where $\pi_b{}'(c,\psi)=c\pi_b(1+\psi^\top g(x,a))$ and $\psi^*$ is defined as

$$\psi^*=\underset{\psi\in\mathbb{R}^{d_\phi}}{\arg\min}\operatorname{var}\left[\left\{\omega(a,x)r-\psi^\top g(x,a)\right\}\right]$$

$$=\mathrm{E}[g(x,a)g(x,a)^\top]^{-1}\mathrm{E}\left[\frac{\pi_e}{\pi_b}rg(x,a)\right].$$

*Proof.* We have

$$\sqrt{n}\left(\mathbb{P}_n\left(\frac{\pi_e r}{\pi_b{}'(\hat{c},\hat{\psi})}\right)-\beta^*\right)$$

$$=\left(\mathbb{G}_n\left(\frac{\pi_e r}{\pi_b{}'(\hat{c},\hat{\psi})}\right)-\mathbb{G}_n\left(\frac{\pi_e r}{\pi_b}\right)\right)+\mathbb{G}_n\left(\frac{\pi_e r}{\pi_b}\right)+\sqrt{n}\left(\mathrm{E}\left[\frac{\pi_e r}{\pi_b{}'(\hat{c},\hat{\psi})}\right]-\beta^*\right)\qquad(12)$$

$$=\mathbb{G}_n\left(\frac{\pi_e r}{\pi_b}\right)+\sqrt{n}\left(\mathrm{E}\left[\frac{\pi_e r}{\pi_b{}'(\hat{c},\hat{\psi})}\right]-\beta^*\right)+o_p(1)\qquad(13)$$

$$=\sqrt{n}\left(\mathbb{P}_n\left(\frac{\pi_e r}{\pi_b}-\psi^* g(x,a)\right)-\beta^*\right)+o_p(1).\qquad(14)$$

From the second line (12) to the third line (13) , noting that $\pi_b{}'(\hat{c},\hat{\psi})$ converges in probability to $\pi_b$ from the fact $\hat{c}\overset{P}{\to}1$ and $\hat{\psi}\overset{P}{\to}(0,0)$ and (a1), (a2), we used:

$$\mathbb{G}_n\left(\frac{\pi_e r}{\pi_b{}'(\hat{c},\hat{\psi})}\right)-\mathbb{G}_n\left(\frac{\pi_e r}{\pi_b}\right)=o_p(1).$$

From the third line (13) to the fourth line (14) , we used the following argument.

$$\sqrt{n}\mathrm{E}\left[\frac{\pi_e r}{\pi_b{}'(\hat{c},\hat{\psi})}\right]=\sqrt{n}\left(\mathrm{E}\left[\nabla_{\psi^\top}\frac{\pi_e r}{\pi_b{}'}\right],\mathrm{E}\left[\nabla_c\frac{\pi_e r}{\pi_b{}'}\right]\right)|_{\psi^*,c^*}((\hat{\psi}-\psi^*)^\top,\hat{c}-c^*)^\top+o_p(1)$$
$$(15)$$

$$=-\mathrm{E}\left[\frac{\pi_e}{\pi_b}rg^\top\right]\mathrm{E}[gg^\top]^{-1}\sqrt{n}\mathbb{P}_n g\qquad(16)$$

$$=\sqrt{n}\mathbb{P}_n[-\psi^{*\top}g].$$

Here, from the first line (15) to the second line (16), we have used the fact that an estimator $\hat{\psi}$ and $\hat{c}$ are defined as an Z-estimator:

$$\mathrm{E}_n\left[\frac{g}{1+\psi^\top g}\right]=0,\ \mathrm{E}_n\left[\frac{1}{1+\psi^\top g}-c\right]=0.$$

This implies

$$\sqrt{n}(\hat{\psi}-\psi^*)=-\mathrm{E}\left[\frac{g(x,a)g(x,a)^\top}{1+\psi^\top g(x,a)}\right]^{-1}\sqrt{n}\mathbb{P}_n g(x,a)|_{\psi^*,c^*}+o_p(1),$$

$$=-\mathrm{E}\left[g(x,a)g(x,a)^\top\right]^{-1}\sqrt{n}\mathbb{P}_n g(x,a)|_{\psi^*,c^*}+o_p(1),$$

$$\sqrt{n}(\hat{c}-c^*)=-\mathrm{E}\left[\frac{g(x,a)}{1+\psi^\top g(x,a)}\right]\sqrt{n}(\hat{\psi}-\psi^*)|_{\psi^*,c^*}+o_p(1)=o_p(1).$$

$$\square$$

Finally, from Lemma D.5, the asymptotic variance of $\hat{\beta}_{\text{emp}}$ is

$$n^{-1} \min_{\psi \in \mathbb{R}^{d_\psi}} \text{var}\left[\{\omega(a,x)r - \psi g(x,a)\}\right].$$

$\square$

*Proof of Theorem 3.3.* We show an asymptotic statement for the practical $\hat{\beta}_{\text{reg}}$ first. Then, we go to the asymptotic statement for the practical $\hat{\beta}_{\text{emp}}$.

We prove the following lemma first.

**Lemma D.6.** $\hat{\zeta} \xrightarrow{p} \zeta^*$, where

$$\zeta^* = \arg\min_{\zeta \in \mathbb{R}^2} \text{E}\left[\{wr - \mathcal{F}(\zeta_1 + \zeta_2 q(x,a;\tau^\dagger))\}^2\right]. \tag{17}$$

*Proof.* We use Theorem 5.7 in [34]. The uniform convergence condition is proved similarly as the proof in Theorem 3.1 based on (a1) and (a2). Therefore, what we have to prove is the minimum of the following function

$$\zeta \to \text{E}\left[\{wr - \mathcal{F}(\zeta_1 + \zeta_2 q(x,a;\tau^\dagger))\}^2\right] \tag{18}$$

is uniquely defined. This is obvious because the above function is a quadratic function with respect to $\zeta$.

$\square$

For the rest of the proof, by following the same argument in the proof of Theorem 3.1 with redefining

$$m^* = \zeta_1^* + \zeta_2^* q(x,a;\tau^\dagger),$$

the statement is proved.

Next, we show a statement for $\hat{\beta}_{\text{emp}}$. As in the proof of Theorem 3.2, we show the following lemma.

**Lemma D.7.**

$$\sqrt{n}\left(\mathbb{P}_n\left(\frac{\pi_e r}{\pi_b'(\hat{c},\hat{\xi},\hat{\tau})}\right) - \beta^*\right) = \sqrt{n}\left(\mathbb{P}_n\left(\frac{\pi_e r}{\pi_b} - \zeta^*(\hat{\tau})g(x,a;\hat{\tau})\right) - \beta^*\right) + o_p(1),$$

*where $\pi_b'(c,\xi,\tau) = c\pi_b(1 + \xi^\top g(x,a;\tau))$ and $\zeta^*(\tau)$ is defined as*

$$\zeta^*(\tau) = \arg\min_{\zeta \in \mathbb{R}^2} \text{var}\left[(\omega(a,x)r - \zeta^\top g(x,a;\tau))\right]$$

$$= \text{E}[g(x,a;\tau)g(x,a;\tau)^\top]^{-1}\text{E}\left[\frac{\pi_e}{\pi_b}rg(x,a;\tau)\right].$$

We go back to the main proof. Finally, we have

$$\sqrt{n}\left(\mathbb{P}_n\left(\frac{\pi_e r}{\pi_b} - \zeta^*(\hat{\tau})^\top g(x,a;\hat{\tau})\right) - \beta^*\right)$$

$$= \mathbb{G}_n\left(\frac{\pi_e r}{\pi_b} - \zeta^*(\hat{\tau})^\top g(x,a;\hat{\tau})\right) - \mathbb{G}_n\left(\frac{\pi_e r}{\pi_b} - \zeta^*(\tau^\dagger)^\top g(x,a;\tau^\dagger)\right)$$

$$+ \mathbb{G}_n\left(\frac{\pi_e r}{\pi_b} - \zeta^*(\tau^\dagger)^\top g(x,a;\tau^\dagger)\right) + \sqrt{n}\left(\text{E}\left[\frac{\pi_e r}{\pi_b} - \zeta^*(\hat{\tau})^\top g(x,a;\hat{\tau})\right] - \beta^*\right)$$

$$= \mathbb{G}_n\left(\frac{\pi_e r}{\pi_b} - \zeta^*(\tau^\dagger)^\top g(x,a;\tau^\dagger)\right) + o_p(1).$$

From the second line to the third line, we use an argument that the first term is equal to $o_p(1)$ by the assumptions (a1), (a2) and the third term is $0$ from the construction. Therefore, the asymptotic variance (MSE) is

$$n^{-1} \min_{\zeta \in \mathbb{R}^2} \text{var}\left[(\omega(a,x)r - \zeta^\top g(x,a;\tau^\dagger))\right].$$

$\square$

*Proof of Theorem 4.1.* We use a law of total variance [2].

$$n\mathrm{var}\left[\mathrm{E}_n\left[\sum_{t=0}^{T-1}\left(\gamma^t\omega_{0:t}r_t - \gamma^t\left(\omega_{0:t}m_t(x_t,a_t) - \omega_{0:t-1}\sum_{a\in A}m_t(x_t,a)\pi_e(a|x_t)\right)\right)\right]\right]$$

$$=\sum_{t=0}^{T-1}\mathrm{E}\left[\mathrm{var}\left(\mathrm{E}\left[\sum_{k=0}^{T-1}\left(\gamma^k\omega_{0:k}r_k - \gamma^k(\omega_{0:k}m_k(x_k,a_k) - \omega_{0:k-1}\sum_{a\in A}m_k\pi_e)\right)|\mathcal{H}_t\right]|\mathcal{H}_{t-1}\right)\right]$$

$$=\sum_{t=0}^{T-1}\mathrm{E}\left[\mathrm{var}\left(\mathrm{E}\left[\sum_{k=t}^{T-1}\left(\gamma^k\omega_{0:k}r_k - \gamma^k(\omega_{0:k}m_k(x_k,a_k) - \omega_{0:k-1}\sum_{a\in A}m_k\pi_e)\right)|\mathcal{H}_t\right]|\mathcal{H}_{t-1}\right)\right]$$

$$=\sum_{t=0}^{T-1}\mathrm{E}\left[\gamma^{2t}\mathrm{var}\left(\mathrm{E}[\sum_{k=t}^{T-1}\gamma^{k-t}\omega_{0:k}r_k|\mathcal{H}_t] - (\omega_{0:t}m_t(x_t,a_t) - \omega_{0:t-1}\sum_{a\in A}m_t\pi_e)|\mathcal{H}_{t-1}\right)\right]$$

$$=\sum_{t=0}^{T-1}\mathrm{E}\left[\gamma^{2t}\omega_{0:t-1}^2\mathrm{var}\left(\mathrm{E}[\sum_{k=t}^{T-1}\gamma^{k-t}\omega_{t+1:k}r_k|\mathcal{H}_t]\omega_{t:t} - (\omega_{t:t}m_t(x_t,a_t) - \sum_{a\in A}m_t\pi_e)|\mathcal{H}_{t-1}\right)\right].$$

From the third line to the fourth line:

$$\mathrm{E}\left[\omega_{0:k}m_k(x_k,a_k) - \omega_{0:k-1}\sum_{a\in A}m_k(x_k,a)\pi_e(a|x_k)|\mathcal{H}_t\right] = 0,$$

for $k > t$. □

*Proof of Lemma 4.1.* Define an estimator as a solution to: $\mathrm{E}_n[d, d_0, d_1, \cdots, d_{T-1}]^\top = 0$, where

$$d = \beta - \left\{\sum_{t=0}^{T-1}\omega_{0:t}\gamma^t r_t/c_t\right\}, \quad d_t = c_t - \omega_{0:t}.$$

The asymptotic MSE of $(\hat\beta, \hat c_1, \ldots, \hat c_{T-1})$ is written as a sandwich formula: $n^{-1}A^{-1}BA^{\top-1}$:

$$A = \begin{pmatrix} 1 & \gamma\beta_1^* & \cdots & \gamma^{T-1}\beta_{T-1}^* \\ 0 & 1 & \cdots & 0 \\ \vdots & \vdots & \ddots & \vdots \\ 0 & 0 & \cdots & 1 \end{pmatrix}, \quad B = \begin{pmatrix} \mathrm{var}[d] & \mathrm{cov}[d,d_1] & \cdots & \mathrm{cov}[d,d_{T-1}] \\ \mathrm{cov}[d_1,d] & \mathrm{var}[d_1] & \cdots & 0 \\ \vdots & \vdots & \ddots & \vdots \\ \mathrm{cov}[d_{T-1},d] & 0 & \cdots & \mathrm{var}[d_{T-1}] \end{pmatrix},$$

where

$$\beta_t^* = \mathrm{E}[\omega_{0:t}r_t].$$

First, $A^{-1}$ is

$$A = \begin{pmatrix} 1 & -\gamma\beta_1^* & \cdots & -\gamma^{T-1}\beta_{T-1}^* \\ 0 & 1 & \cdots & 0 \\ \vdots & \vdots & \ddots & \vdots \\ 0 & 0 & \cdots & 1 \end{pmatrix}.$$

Then, the (1,1) element in $A^{-1}BA^{\top-1}$ is

$$\mathrm{var}[d] - \sum_{t=0}^{T-1}\gamma^t\beta_t^*\mathrm{cov}[d,d_t] + \sum_{t=0}^{T-1}\gamma^{2t}\beta_t^{*2}\mathrm{var}[d_t]. \tag{19}$$

First, $\mathrm{var}[d]$ is equal to

$$\sum_{t=0}^{T-1}\mathrm{E}\left[\gamma^{2t}\omega_{0:t-1}^2\mathrm{var}\left(\mathrm{E}\left[\sum_{k=t}^{T-1}\gamma^{k-t}\omega_{t+1:k}r_{k-t}|\mathcal{H}_t\right]\omega_{t:t}|\mathcal{H}_{t-1}\right)\right].$$

Then, $\text{cov}[d, d_t]$ is equal to

$$\text{E}\left[\gamma^{2k}\omega_{0:t-1}^2\text{cov}\left(\text{E}\left[\sum_{k=t}^{T-1}\gamma^{k-t}\omega_{t+1:k}r_{k-t}|\mathcal{H}_t\right]\omega_{t:t}, \beta_t^*\omega_{t:t}|\mathcal{H}_{t-1}\right)\right].$$

Finally, the term (19) is equal to

$$\sum_{t=0}^{T-1}\text{E}\left[\gamma^{2t}\omega_{0:t-1}^2\text{var}\left(\omega_{t:t}\left(\text{E}[\sum_{k=t}^{T-1}\gamma^{k-t}\omega_{t+1:k}r_{k-t}|\mathcal{H}_t] - \beta_t^*\right)|\mathcal{H}_{t-1}\right)\right].$$

$\square$

*Proof of Theorem 4.2.* As in the same way of Theorem 3.1, it is proved that the asymptotic MSE of $\hat{\beta}_{\text{reg}}^{T-1}$ is

$$n^{-1}\min_{\zeta\in\mathbb{R}^{d_\zeta}}\text{var}[v(\{\zeta_{1t} + \zeta_{2t}q(x, a; \tau^\dagger)\}_{t=0}^{T-1})].$$

We prove the intrinsic efficiency. Regarding local efficiency, they are proved as the proof of Corollary 3.1. The asymptotic MSEs of $\hat{\beta}_{\text{sis}}$, $\hat{\beta}_{\text{snsis}}$ and $\hat{\beta}_{\text{dr}}$ are represented as a form of $n^{-1}\text{var}[v(\{\zeta_{1t} + \zeta_{2t}q(x, a; \tau^\dagger)\}_{t=0}^{T-1})]$. Setting $\zeta_{1t} = 0$ and $\zeta_{2t} = 0$, it corresponds to the estimator $\hat{\beta}_{\text{sis}}$. Setting $\zeta_{1t} = \beta_t^*$ and $\zeta_{2t} = 0$, it corresponds to the estimator $\hat{\beta}_{\text{snsis}}$. Setting $\zeta_{1t} = 0$ and $\zeta_{2t} = 1$, it corresponds to the estimator $\hat{\beta}_{\text{dr}}$. This concludes the intrinsic efficiency.

$\square$

*Proof of Theorem 4.3.* We prove a 1-boundedness and (partial) stability. When $\tau$ is not pre-estimated, it has stability. When $\tau$ is pre-estimated, it has partial stability. We prove the latter point. Regarding the asymptotic result, we can prove as in Theorem 3.2.

From the first consider of optimization problem with respect to $\zeta_{1t}$ for $0 \leq t \leq T-1$, we have

$$0 = \text{E}_n\left[\frac{w_{0:t} - w_{0:t-1}}{1 + g(\mathcal{D}_{x,a}; \hat{\xi}, \hat{\tau})}\right].$$

Noting $w_{0:-1} = 1$ for any $t$,

$$0 = \text{E}_n\left[\frac{w_{0:t} - 1}{1 + g(\mathcal{D}_{x,a}; \hat{\xi}, \hat{\tau})}\right]. \tag{20}$$

The estimator $\hat{\beta}_{\text{emp}}^{T-1}$ is bounded as follows. Regarding the 1-boundedness,

$$\hat{\beta}_{\text{emp}}^{T-1} \leq \frac{1}{n}\sum_{i=1}^{n}\sum_{t=0}^{T-1}\omega_{0:t}^{(i)}\gamma^t r_t^{(i)}\frac{\hat{c}^{-1}}{1 + g(\mathcal{D}_{x,a}; \hat{\xi}, \hat{\tau})}$$

$$\leq \frac{1}{n}\sum_{i=1}^{n}\sum_{t=0}^{T-1}\omega_{0:t}^{(i)}\gamma^t R_{\max}\frac{\hat{c}^{-1}}{1 + g(\mathcal{D}_{x,a}; \hat{\xi}, \hat{\tau})}$$

$$= \sum_{t=0}^{T-1}\gamma^t R_{\max}$$

From the second to the third line, we have used (20).

Regarding the partial stability, noting that from the assumption, $\hat{\zeta}$ and $\hat{\tau}$ are functions of $\mathbf{x}$ and $\mathbf{a}$,

$$\mathrm{var}[\hat{\beta}_{\mathrm{emp}}^{T-1}|\mathcal{D}_{x,a}] \leq \mathrm{var}\left[\frac{1}{n}\sum_{i=1}^{n}\sum_{t=0}^{T-1}\omega_{0:t}^{(i)}\gamma^{t}r_{t}^{(i)}\frac{\hat{c}^{-1}}{1+g(\mathcal{D}_{x,a};\hat{\xi},\hat{\tau})}\bigg|\mathcal{D}_{x,a}\right]$$

$$\leq \frac{1}{n}\sum_{i=1}^{n}\sum_{t=0}^{T-1}\left\{\omega_{0:t}^{(i)}\frac{\hat{c}^{-1}}{1+g(\mathcal{D}_{x,a};\hat{\xi},\hat{\tau})}\right\}^{2}\gamma^{2t}\mathrm{var}[r_{t}]$$

$$\leq \frac{1}{n}\sum_{i=1}^{n}\sum_{t=0}^{T-1}\left\{\omega_{0:t}^{(i)}\frac{\hat{c}^{-1}}{1+g(\mathcal{D}_{x,a};\hat{\xi},\hat{\tau})}\right\}^{2}\gamma^{2t}\max[\mathrm{var}[r_{t}]]$$

$$\leq \sum_{t=0}^{T-1}\gamma^{2t}\max[\mathrm{var}[r_{t}]] = \sigma^{2}.$$

From the third to the fourth line, we have used (20).

$\square$

*Proof of Theorem B.1.* The estimator is defined as a solution to the following equation with respect to $\beta$, $c$:

$$\mathrm{E}_{n}[d_{1},d_{2}] = 0,$$

where

$$d_{1}(x,a;\beta,c) = \beta - \frac{\omega_{0:0}(x,a)}{c}(r-m(x,a)) - \left\{\sum_{a\in A}m(x,a)\pi_{e}(a|x)\right\}, \quad d_{2}(x,a;c) = c - \omega_{0:0}(x,a).$$

The asymptotic MSE of $(\hat{\beta},\hat{c})$ is written as

$$\mathrm{Asmse}[(\beta,c)^{\top}] = \begin{bmatrix}1 & \mathrm{E}[\nabla_{c}d_{1}] \\ 0 & 1\end{bmatrix}^{-1}\begin{bmatrix}\mathrm{var}[d_{1}] & \mathrm{cov}[d_{1},d_{2}] \\ \mathrm{cov}[d_{1},d_{2}] & \mathrm{var}[d_{2}]\end{bmatrix}\begin{bmatrix}1 & 0 \\ \mathrm{E}[\nabla_{c}d_{1}] & 1\end{bmatrix}^{-1}\bigg|_{\beta^{*},c^{*}}$$

$$= \begin{bmatrix}1 & -\mathrm{E}[\nabla_{c}d_{1}] \\ 0 & 1\end{bmatrix}\begin{bmatrix}\mathrm{var}[d_{1}] & \mathrm{cov}[d_{1},d_{2}] \\ \mathrm{cov}[d_{1},d_{2}] & \mathrm{var}[d_{2}]\end{bmatrix}\begin{bmatrix}1 & 0 \\ -\mathrm{E}[\nabla_{c}d_{1}] & 1\end{bmatrix}\bigg|_{\beta^{*},c^{*}}.$$

Therefore, the asymptotic MSE is given as

$$(\mathrm{var}[d_{1}] - 2\mathrm{E}[\nabla_{c}d_{1}]\mathrm{cov}[d_{1},d_{2}] + \mathrm{E}[\nabla_{c}d_{1}]^{2}\mathrm{var}[d_{2}])|_{\beta^{*},c^{*}}.$$

Here, noting that $c^{*}=1$,

$$\mathrm{E}[\nabla_{c}d_{1}]|_{c^{*}} = \mathrm{E}\left[\omega(a,x)(r-m(x,a))\right],$$

$$\mathrm{cov}[d_{1},d_{2}]|_{c^{*}} = \mathrm{E}\left[\omega_{0:0}^{2}(a,x)(r-m(x,a)) - \omega(a,x)\left\{\sum_{a\in A}\pi_{e}(a|x)m(x,a)\right\}\right] - \beta^{*},$$

$$\mathrm{var}[d_{1}]|_{\beta^{*},c^{*}} = \mathrm{var}\left[\omega(a,x)(r-m(x,a)) - \left\{\sum_{a\in A}\pi_{e}(a|x)m(x,a)\right\}\right],$$

$$\mathrm{var}[d_{2}]|_{\beta^{*},c^{*}} = \mathrm{var}\left[\omega(a,x)\right].$$

By combining all together, we get the conclusion.

Note that the influence function is written as

$$d_{1}(x,a)|_{\beta^{*},c^{*}} - \mathrm{E}[\nabla_{c}d_{1}(x,a)]|_{c^{*}}d_{2}(x,a)|_{\beta^{*},c^{*}}. \tag{21}$$

$\square$

*Proof of Theorem B.2.* Define $u_{b}(c,m(x,a;\zeta,\tau))$:

$$\beta = \sum_{a\in A}m(x,a;\zeta,\tau)\pi_{e}(a|x) + \frac{\omega_{0:0}(x,a)}{c}(r-m(x,a;\zeta,\tau)).$$

By noting that $\hat{c} \xrightarrow{p} c^* = 1$, this is decomposed as

$$\sqrt{n}(u_b(\hat{c}, \hat{m}) - \beta^*) = \mathbb{G}_n[u_b(\hat{c}, \hat{m})] - \mathbb{G}_n[u_b(1, m^*)]$$
$$+ \mathbb{G}_n[u_b(1, m^*)]$$
$$+ \sqrt{n}(\mathrm{E}[u_b(\hat{c}, \hat{m})] - \beta^*),$$

when $m^* = \zeta_1^* + \zeta_2^* q(x, a; \tau^*)$ and $\hat{m} = \hat{\zeta}_1 + \hat{\zeta}_2 q(x, a; \hat{\tau})$. Here, the first term is equal to $\mathrm{o}_p(1)$ from assumptions (a1) and (a2). The last term is

$$\sqrt{n}(\mathrm{E}[u_b(\hat{c}, \hat{m})] - \beta^*)$$
$$= \sqrt{n}(\mathrm{E}[u_b(1, \hat{m})] - \beta^*) + \sqrt{n}\mathrm{E}[\nabla_c u_b(c, m)]|_{c^*} \mathbb{P}_n \left( \omega(a, x) - 1 \right) + \mathrm{o}_p(1)$$
$$= \sqrt{n}\mathrm{E}[\nabla_c u_b(c, m)]|_{c^*} \mathbb{P}_n \left( \omega(a, x) - 1 \right) + \mathrm{o}_p(1).$$

Therefore,

$$\sqrt{n}(u_b(\hat{c}, \hat{m}) - \beta^*) = \mathbb{G}_n \left[ u_b(1, m^*) + \mathrm{E}[\nabla_c u_b(c, m)]|_{c^*} \left( \omega(a, x) - 1 \right) \right] + \mathrm{o}_p(1).$$

From the form of the influence function (21), this implies that the asymptotic MSE is

$$n^{-1} \min_{\zeta \in \mathbb{R}^2, \tau \in \Theta_\tau} V_{\mathrm{snd}}(m(x, a; \zeta, \tau)).$$

$\square$

*Proof of Theorem C.1.* As in the same way of Theorem 3.1, it is proved that the asymptotic MSE of $\hat{\beta}_{\mathrm{reg}}^0$ is

$$n^{-1} \min_{\zeta \in \mathbb{R}^2, \tau \in \Theta_\tau} \mathrm{var}[v(\{\zeta_1 + \zeta_2 q(x, a; \tau)\}_{t=0}^{T-1})].$$

The asymptotic MSE of $\hat{\beta}_{\mathrm{sis}}$, $\hat{\beta}_{\mathrm{sn2reg}}$ and $\hat{\beta}_{\mathrm{dr}}$ is represented as a form of $\mathrm{var}[v(\{\zeta_1 + \zeta_2 q(x, a; \tau)\}_{t=0}^{T-1})]$. When $\zeta = (0, 0)$, it corresponds to the $\hat{\beta}_{\mathrm{sis}}$. When $\zeta = (\beta^*, 0)$, it corresponds to the $\hat{\beta}_{\mathrm{sn2sis}}$. When $\zeta = (0, 1)$, it corresponds to $\hat{\beta}_{\mathrm{sndr}}$. $\square$

# E   Details of the experimental setup

## E.1   Contextual bandit

**Transformation method** A multi-label classification data set comprises $(x^{(i)}, y^{(i)})_{i=1}^n$ where $x^{(i)}$ is covaraite and $y^{(i)}$ is its class. Here, $x^{(i)} \in \mathbb{R}^d$ and $y^{(i)} \in \{1, \cdots, l\}$, where $l$ is the number of class. A classification algorithm assigning $x$ to $y$ is considered to be a policy from a context to an action.

Next, we will explain how to define a reward. The policy is considered to be an estimator $a^{(i)}$ associated with $x^{(i)}$. The agent receives a unit reward $1$ if the prediction succeeds, that is, when $a^{(i)} = y^{(i)}$. It receives no reward when $a^{(i)} \neq y^{(i)}$. The reward of a policy is considered to be the accuracy of the classification model. In this way, we can generate triplets of $\{(x^{(i)}, a^{(i)}, r^{(i)})\}_{i=1}^n$. In section 5, based on some classification data set and some randomized policies, we made a data set 200 times and performed simulations.

**Additional remarks**

- The data set is split into training data (30%) for defining a policy $\pi_d$ and evaluation data (70%) for the OPE. The size of the evaluation data is larger than that of training data because for the current problem, the accuracy of $\pi_d$ is not important and, we intend to know the accuracy of the OPE methods.

- Our survey has denoted that several methods can be employed to construct Q-functions. For example, [6] used a training data set to learn a Q-function. However, we should not use the training data for the valid comparison of OPE methods. In our case, the behavior policy was only applied to the evaluation data set. We subsequently constructed a Q-function using the generated data.

- The number of actions and data points of the problem is shown in Table 12.

Table 12: Bandit Datasets

| Dataset | PageBlock | OptDigits | SatImage | PenDigits |
|---------|-----------|-----------|----------|-----------|
| Classes | 5 | 10 | 6 | 10 |
| Data | 5473 | 5620 | 6435 | 10992 |

## E.2 Reinforcement learning

We here describe the RL domains used in the experiments. In addition, we show the graph representing the results in Table 2.

**Windy Gridworld**

A detailed explanation is Example 6.5 in [27]. The board is a $7 \times 10$ matrix. The reward is $-1$ for all tranistion until the terminal state is reached. The action comprises four choices: up, down, right, left. The difference of the usual GridWorld is that a crosswind runs upward through the middle of the grid. The horizon was set to $T = 400$. Further, we calculated the best policy $\pi_d$ using Q-learning.

**Cliff Walking**

The detailed explanation is Example 6.6 in [27]. The board is a $4 \times 12$ matrix. Each time step incurs $-1$ reward, and stepping into the cliff incurs $-100$ reward and a reset to the start. An episode is terminated when the agent reaches the goal. The horizon was set to $T = 400$. Further, we calculated the best policy $\pi_d$ using Q-learning.

**Mountain Car**

A car is between two hills in interval $[-0.7, 0.5]$ and the agent should move back and forth to gain enough power to reach the top of the right hill. The state space comprises position and velocity. There are three discrete actions 1)forward, 2)backward and 3) stay-still. The horizon was set to be $T = 250$ with a reward of $-1$ per step. We calculated the best policy $\pi_e$ using Q-learning. The state space was continuous; thus, we obtained a 400-dimensional feature using a radial basis function kernel.

Figure 2: RL results; Cliff Walking and Mountain Car