[Reviews · NeurIPS 2019]

Reviewer 1



4) Originality: The background is extensive and clearly position the proposed methods in relation to the previous techniques used in the field. The proposed estimators are a combination of well-known estimators. The obtained estimators achieve boundedness and stability properties of SNIS and the consistency and local efficiency of DR methods. To my knowledge, the proposed estimators are the only estimators having all these properties. 5) Clarity: The math of this submission is fairly involved, but it remains accessible for non-experts. The submission is clearly written and introduced every definition needed to understand the contribution. The notation is consistent and not overly complex. 6) Quality: The contribution is technically sound. The results for the RL experiments show good improvements over the baselines. 7) Significance: I believe that the methods proposed in this work are likely to be built upon by the community. Specifically, off-policy RL methods rely extensively on OPE methods and such an improvement could lead to more efficient algorithms. Conclusion: I believe this work should be accepted given all the properties of the proposed estimators that were not satisfied by existing estimators. Furthermore, the OPE results on CB and particularly RL are convincing.

Reviewer 2



Originality: This work is based on the MDR estimator for OPE. By modifying MDR and changing the way of choosing the parameters, the proposed estimators can simultaneously meet all the desired properties defined in the paper (i.e., efficiency, stability, and boundedness.). It is a straightforward and effecient idea, although a bit incremental. Quality: The technical points claimed in this paper make sense, also with theoretical guaranttees. However, I have a few questions about the experimental settings. 1) In the CB/RL experiments why do the authors choose those evaluation/behavoiur policies rather than the others? Will different choices lead to different results/conclusions? 2) From the tables in both CB/RL experiments, I did not see some obvious difference between DR and REG/EMP, particularly in Table 3, 4 & 7. Otherwise, how can we judge that the minor differences between them come from the algorithms but not other factors (e.g., the choice of behaviour policies, measurement error, or even variance)? 3) In all the tables, what are the variances of all the approaches in terms of RMSE, which I think is a good indicator as well. Clarity: Generally the paper is organized in a clear manner and is well written, but at times there exist some notational confusion and unclear statements. For example, the author did not explain what MDR stands for in the paper although providing the reference. On Line 60, the authors claim that "all of models are wrong to some extent" but did not say why and in which extent, etc. Significance: Although the proposed method has a reasonble guaranttee in theory, I am still in doubt how to use it in real-world RL applications. Because what I saw is that they used the linear/logistic function in all the experiments, which makes no sense in many RL problems, even in many RL simulations.

Reviewer 3



Updated comments : I confirm I have read the author response. I would appreciate, if the paper is accepted, that the authors include more clarity in explaining their construction and approach. As authors have pointed, it would be useful to see more clearly, how in the context of off-policy evaluation, normalized empirical likelihood makes a difference. Empirical results : I think the paper needs more work in clearly demonstrating their empirical results. As authors have stated, it would be great to see the plots of their results table, and code being made available upon acceptance. This work would be useful to the community in general, especially as the authors portrays their proposed method in light of other estimators such as doubly robust. Such a contribution would be useful to the community. I do agree, and appreciate the reasons why the authors first demonstrated their results in context of contextual bandits, and then extended it to the RL case. More clarity in writing upon acceptance of the paper would be useful. ------------------------------------------------------- Comments : _x000B_ - The paper provides a good summary and puts into context much of the off-policy policy evaluation estimators, where different approaches have been proposed previously to reduce variance while providing unbiased estimates. This is a key study in the OPE literature, as methods to provide better stability for off-policy methods are required for practical applications of RL. _x000B_ - Table 1 is useful - provides a good summary and comparison of existing OPE estimators. Section 2.1 further provides a good summary of existing OPE estimators based on consistency, stability and boundedness. This is well written and easy to follow - and useful for the community as it provides a direct comparison between existing OPE estimators in terms of several properties. _x000B_ - The key idea of the paper is then to propose a parameterized class of estimators that satisfy the best of both DR and SNIS based OPE estimators. The authors argue that the proposed estimator can achieve desired properties in terms of boundedness, stability, local efficiency, whereas none of the existing estimators can achieve all of the these desired properties. The proposed estimator is parameterized and includes a combination of IS and DR estimators - from which either IS or DR estimators can be retrieved by setting one of the weights to 0. Section 3 and 4 shows the boundedness and efficiency of the proposed estimator for contextual bandits and RL settings respectively. _x000B_ - Sections 3 and 4 are a bit hard to follow - I suggest that in the final version of the paper, the authors make more efforts into presenting the desired properties in a more intuitive and explanable way, instead of simply presenting how the proposed estimator can theoretically achieve the desired properties. _x000B_ - The authors, however, do a great job in proposing their novel contribution - that a parameterized estimator can be used where the parameters can be estimated with likelihood instead of MSE. The authors then go on to derive the likelihood estimator (above theorem 4.3, non-labeled eqn) and show that the empirical policy value achieves the same as MSE asympotically. _x000B_ - Empirical results with the proposed estimator are presented for both contextual bandits and RL - especially for the RL setting, it can be seen clearly that the proposed estimator achieves lower RMSE compared to all the proposed baselines. _x000B_ - It would have been useful if the code for these experiments were provided - or if results were provided in a figure too (instead of tabular results). While I agree that the proposed estimator seems to be useful across all the presented tasks - in terms of empirical variance and stability, it would have been useful to see actual figures of the RMSE minimizing to these values. I suggest the authors provide more extensive results in the final version of the paper. - Overall, the paper provides an important step in off-policy policy evaluation - proposing new classes of estimators which can combine the best of existing estimaotrs in the OPE literature. The authors provide an extensive proof and detail of the properties of their proposed estimator (including in the appendix) and empirical results seem to suggest that the proposed estimator can achieve better performance in terms of RMSE compared to several baselines widely used in OPE. The authors provide results for both CB and RL setting in terms of RMSE - although it would have been useful to see actual figures of the RMSE for all these tasks (either in the main or in appendix) to clearly highlight how the proposed estimator can outperform baselines such as SNIS and DR estimators. The paper is a bit hard to follow, with the results not often presented in a very clear or concise way, and includes a lot of details of the desired properties - however the authors do a great job in putting into context how the proposed estimators can be better and includes a detailed discussion of existing OPE methods in the literature. _x000B_ - If the above comment can be highlighted, and the paper written in a more clear and descriptive way - I would vote for acceptance for this paper as it provides significantly novel contribution in existing literature of off-policy methods.

[Author Response · NeurIPS 2019]

We thank the reviewers for their useful feedback. Our responses are below. We mistakenly omitted the code from
supplemental material at submission, but it will be public following the new NeurIPS rules for accepted papers (code is
already on GitHub in fact) to enable reproduction and extensions of our work, and to allow others to try a greater suite
of experiment settings. Unfortunately, links (even anonymous) are disallowed in the response.

**Reviewer 1:**

8) Thanks for catching. Will fix typo.

9) We will add more detail. The figure seeks to illustrate the local and intrinsic efficiency uniquely achieved by our
new estimators. It shows the ordering of asymptotic MSEs: if Q-functions are well specified, both our proposed
estimators (EMP, REG) and DR achieve the same efficiency bound, which IS and SNIS do not; if Q-functions
are misspecified, the efficiency bound is not achieved, yet our proposed estimators will *still* have better MSE
than DR, IS, and SNIS.

10) Thank you for the great reference. We will definitely cite it.

**Reviewer 2:**

1) We followed Farajtabar et al. [5] when constructing evaluation and behavior policies. Our policies are the ones
that they call friendly softening in their paper. We tried experiments on some other behavior and evaluation
policy cases, and the overall results were the same (when policies are close, DR performs well among baselines
and we match/exceed it; when policies are far, SNIS performs well among baselines and we match/exceed it).
We will include the additional results in the supplement. Additionally, the availability of the code allows one to
easily play with the policy parameters.

2) Firstly, the difference is small but usually statistically significant; we will add standard errors. Second, Tables
2–4 show that the our estimators (EMP, REG) are competitive with DR in the settings where it works, and beat it
handily when it does not work, such as when the behavior and evaluation policies differ as in the Tables' 3rd
line. We will comment. Similarly, Tables 5–7 show a variety of settings. While DR is sometimes competitive
and sometimes less so, our estimators perform well throughout. We will add a comment that in some settings
many estimators perform similarly well (DR, MDR, and ours) while in others we outperform. The point is we
are never worse and sometimes better. This verifies our theory.

3) We will add standard errors.

**Clarity**:

1) Good point. We will add "More Robust Doubly Robust" at the first mention of MDR.

2) Re L60, we just meant to highlight that local efficiency is limited because correct parametric specification is a
strong assumption. We will clarify.

**Significance**: Even if the Q-model is a neural network, our proposed method still works and our guarantees will
still hold. We agree that often using such Q-models is necessary in more complicated situations. However, for the
experiments studied, which are standard in the literature we build on, all of the domains are not complex and we
followed the previous literatures regarding the choice of Q-model to most clearly demonstrate the contribution of
our approach relative to previous work.

**Reviewer 3:**

• Re Sec 3,4 clarity: we will follow your suggestion and add more intuition on the construction. Sec 3 follows
similar ideas to [5], whereas Sec 4 uses a new modified (normalized) empirical likelihood approach, which we
will explain and connect more clearly to standard empirical likelihood.

• Re code: apologies for the omission in supplement submission; the code will be public. See above.

• Plots of the tables will be added to the supplement. Obviously, there were space constraints. See also R2(2)
about additional discussion of results.

• Re improvements:

– See above re Sec 3,4. Additionally, we will do another editing pass to improve clarity.
– There are many results and we wanted to clearly contextualize the contribution in the literature. We tried
to add clarity by first discussing the CB case even though it is just a special case of the RL setting. We
will add additional intuitive explanations in Sec 3.
– Code will definitely be provided for reproducible research and to allow others to build on our work.

[Meta-Review · NeurIPS 2019]

This paper presents new estimators for Off Policy Evaluation (OPE) based on likelihoods and argues that the new estimators are better than Importance Sampling (IS). The paper provides strong theoretical guarantees of the estimators, and demonstrates their through simple experiments. The reviewers agree that the paper is well written overall and the proposed methods are technically sound and likely to be built upon by the community. One reviewer is unsure if the proposed methods will be practical in RL applications. The experiments are performed on very simple tasks. The authors should improve this particular point to make a good impact.